# A new large hypercarnivorous crocodyliform from the Maastrichtian of Southern Patagonia, Argentina

**Fernando E. Novas**[1,2,3]**, Diego Pol** ⓘ[2,4]*****, Federico L. Agnolín**[1,2,3]**,
Ismar de Souza Carvalho** ⓘ[5,6]**, Makoto Manabe**[7]**, Takanobu Tsuihiji**[7]**,
Sebastián Rozadilla**[1,2]**, Gabriel L. Lio**[1]**, Marcelo P. Isasi**[1,2,3]

**1** Laboratorio de Anatomía Comparada y Evolución de los Vertebrados (LACEV) Museo Argentino de Ciencias Naturales "Bernardino Rivadavia" (MACN), Buenos Aires, Argentina, **2** Consejo Nacional de Investigaciones Científicas y Técnicas (CONICET), Buenos Aires, Argentina, **3** Fundación de Historia Natural "Félix de Azara", Centro de Ciencias Naturales, Ambientales y Antropológicas, Universidad Maimónides, Buenos Aires, Argentina, **4** Museo Argentino de Ciencias Naturales "Bernardino Rivadavia" (MACN), Buenos Aires, Argentina, **5** Universidade Federal do Rio de Janeiro, Rio de Janeiro, RJ, Brazil, **6** Centro de Geociências, Universidade de Coimbra, Coimbra, Portugal, **7** Department of Earth and Planetary Science, The University of Tokyo, Bunkyo-ku, Tokyo, Japan

* cacopol@gmail.com

## Abstract

The first crocodyliform specimen from the Maastrichtian Chorrillo Formation (Austral Basin, Patagonia) is here described. The discovery was made about 30 km to the SW of the town of El Calafate (Province of Santa Cruz, Argentina) and consists of a beautifully preserved and articulated skull and jaws, and part of the postcranial skeleton that were preserved encased in a large concretion. This new taxon belongs to the notosuchian clade Peirosauridae, representing the latest and southernmost record for this group of crocodyliforms. The new taxon is recovered as closely related to other robust and broad-snouted peirosaurids that lived by the end of the Cretaceous Period, such as *Colhuehuapisuchus* from the Maastrichtian of Central Patagonia and *Miadanasuchus oblita* from the Maastrichtian of Madagascar. The completeness of the new specimen reveals, for the first time, the anatomy and body plan of a large and broad snouted peirosaurid. The new taxon bears large ziphodont teeth, a broad oreinirostral snout that is only slightly longer than 50% the skull length, and a deep adductor chamber in the temporal region and posterior mandibular ramus. The anterior region of its postcranial skeleton is preserved and shows broad scapula and a robust humerus features previously known in large predatorial notosuchians. The new crocodyliform adds to the predatorial component of terrestrial ecosystems at high paleolatitudes by the end of the Cretaceous Period.

**Data availability statement:** all relevant data are within the paper and its Supporting Information files.

**Funding:** DP 9282-R-22 National Geographic Society https://www.nationalgeographic.org/society/ The funders had no role in study design, data collection and analysis, decision to publish, or preparation of the manuscript. ISC Faperj E-26/200.998/2024 Fundação Carlos Chagas Filho de Amparo à Pesquisa do Estado do Rio de Janeiro https://www.faperj.br/ The funders had no role in study design, data collection and analysis, decision to publish, or preparation of the manuscript. ISC CNPq 303596/2016-3 Conselho Nacional de Desenvolvimento Científico e Tecnológico https://www.gov.br/cnpq/pt-br The funders had no role in study design, data collection and analysis, decision to publish, or preparation of the manuscript.

**Competing interests:** The authors have declared that no competing interests exist.

## Introduction

The end of the Cretaceous Period is particularly well recorded in Patagonia [1,2]. In recent years a large collecting effort has resulted in a high number of new vertebrate records in different basins of Patagonia, including the discovery of new and diverse faunal associations in central [e.g., 3,4] and southern [e.g., 5–7] Patagonia that complements previous knowledge from the Neuquén Basin in northern Patagonia [e.g., 1,8–11]. Among the records from the Austral Basin, the most noteworthy are those from the Campanian Cerro Fortaleza [e.g., 6] and the Maastrichtian Chorrillo formations [e.g., 5,12] in Argentina, and Cerro Dorotea Formation in southern Chile [e.g., 7,13].

The Chorrillo Formation crops out in the SW corner of Santa Cruz Province, in Argentine Patagonia (Fig 1). These beds are particularly interesting as they have yielded a wide array of fossils from a Maastrichtian terrestrial ecosystem [5,12], including pollen and spores [14], plant remains [15], freshwater and terrestrial invertebrates [16], fishes, frogs, and turtles. Dinosaurs recovered from these beds include the elasmarian *Isasicursor santacrucensis* [5,17,18], the large titanosaur *Nullotitan glaciaris* [5], the megaraptorid coelurosaur *Maip macrothorax* [18], the birds *Kookne yeutensis* and *Yatenavis iujensis* [5,19], as well as indeterminate parankylosaurs and hadrosaurids [17]. The vertebrate fossil assemblage also includes representatives of different mammalian clades, including monotremes, gondwanatherians, meridiolestidans, and therians [20–22].

Here, we expand the diversity of fossil vertebrates from the Chorrillo Formation with the description of the peirosaurid crocodyliform *Kostensuchus atrox* nov. gen. et sp., represented by an exquisitely preserved and articulated skeleton, lacking some of the limb bones and tail (S1 File). This specimen is one of the best preserved and anatomically informative peirosaurid crocodyliform yet recorded, and the most complete representative of robust, broad snouted members of this clade. The new taxon was a large predator, approximately 3.5 meters long (estimation based on extrapolations with complete skeletons of *Caiman* and *Alligator*), second in size, among predators, only to the megaraptorid theropod *Maip* (ca. 9 meters) [18] known from the Chorrillo Formation. *Kostensuchus* reveals the craniomandibular anatomy and body plan of the large and robust peirosaurids that survived until the end Cretaceous in South America and Madagascar, and provides the southernmost record of this diverse crocodyliform clade. The record of *Kostensuchus* contributes to characterize the faunal association from the Maastrichtian of southern Patagonia (Austral Basin), and underscores the differences among coeval terrestrial ecosystems in central and northern Patagonia (Cañadón Asfalto and Neuquén basins).

## Methods

Body mass was estimated based on dorsal cranial length (490 mm), following the approach of recent studies for notosuchian crocodyliforms [23] and the scaling equations based on extant crocodylians [24].

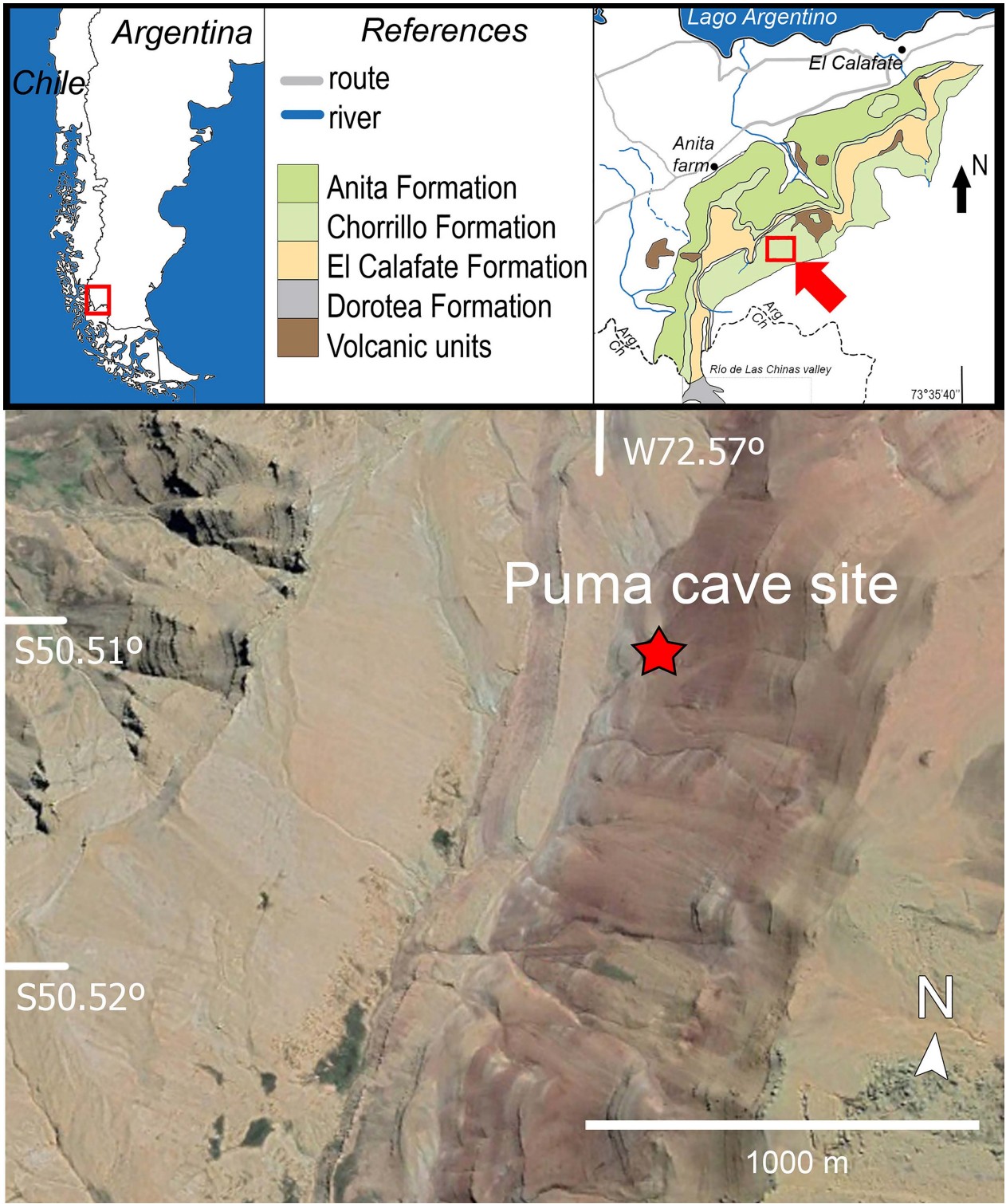

**Fig 1. Map of the fossil locality of *Kostensuchus* gen. nov.** The map shows the locality in southern Patagonia (Santa Cruz Province, Argentina). Modified from [5].

The affinities of *Kostensuchus* were tested using an expanded version taken from Fernández-Dumont et al. [25], which is in turn a modified version of datasets previously published [26,27] in which we added the new taxon and increased the sample of peirosaurid crocodyliforms. The complete dataset includes 120 taxa scored across 444 characters and was analyzed using an equally weighted parsimony analysis in TNT v. 1.6 [28]. Tree searches consisted of using New Technology Searches until 100 hits to minimum length was achieved. A subsequent round of TBR branch swapping was applied on the most parsimonious trees obtained. Unstable taxa were detected using IterPCR [29,30] and nodal support was evaluated using parsimony jackknife [31] considering the impact of unstable taxa [32].

The specimen was found in a concretion and mechanically prepared (S1 File) and a ll necessary permits were obtained for the described study, which complied with all relevant regulations.

**Institutional abbreviations.** CPPLIP, Centro de Pesquisas Paleontológicas Llewellyn Ivor Price, Peirópolis, Minas Gerais, Brazil; MPM, Museo Provincial Padre Molina, Río Gallegos, Santa Cruz, Argentina; ROM, Royal Ontario Museum, Montreal, Canada.

## Nomenclatural acts

The electronic edition of this article conforms to the requirements of the amended International Code of Zoological Nomenclature, and hence the new names contained herein are available under that Code from the electronic edition of this article. This published work and the nomenclatural acts it contains have been registered in ZooBank, the online registration system for the ICZN. The ZooBank LSIDs (Life Science Identifiers) can be resolved and the associated information viewed through any standard web browser by appending the LSID to the prefix ""http://zoobank.org/"".

The LSID for this publication is:

urn:lsid:zoobank.org:pub:D78984D3-64A3-4EEF-AA80-1D5C366D04AB

The electronic edition of this work was published in a journal with an ISSN, and has been archived and is available from the following digital repositories: PubMed Central, LOCKSS.

## Results

### Systematic paleontology

Crocodyliformes Hay, 1930

Mesoeucrocodylia Whetstone and Whybrow, 1980

Peirosauria Leardi et al., 2024

Peirosauridae Gasparini, 1982

*Kostensuchus atrox* gen. et sp. nov.

urn:lsid:zoobank.org:act:4A3D540F-FBE4–4581-97D4-9F6ED0F793D4

**Etymology.** *Kosten*, refers to the Patagonian wind in Aonikenk language; and *suchus*, latinized from the Greek Souchos in references to the Egyptian crocodile-headed god (Sebek). The species epithet *atrox* means harsh in Greek.

**Holotype.** MPM-PV 23554, articulated skull, jaws, cervical, axial skeleton remains including cervical, dorsal, and sacrals, rows of osteoderms along the vertebral sequence, cervical and dorsal ribs, scapular and pelvic girdles, complete right humerus and partial left humerus, and fragmentary remains of the hindlimbs.

**Locality, horizon, and age.** Lower section of the Chorrillo Formation, at Estancia La Anita, approximately 30 km SW from El Calafate city (Fig 1). The specimen comes from a concretionary level exposed at the locality "Puma cave" (locality 4, 5), approximately 60 meters from the base of the Chorrillo Formation [12].

**Diagnosis.** *Kostensuchus* gen. nov. is among the largest known peirosaurids (dorsal skull length = 49 cm) diagnosed by the following combination of characters (autapomorphies marked by *): skull proportionally shorter, wider and higher than in other peirosaurids (e.g., *Hamadasuchus*, *Lomasuchus*, *Montealtosuchus*); rostrum comprising slightly over 50%

the total skull length; sinusoidal alveolar margin of maxilla (shared with *Hamadasuchus*); completely ossified internarial bar; lacrimal lateral surface faintly ornamented and slightly depressed between antorbital fenestra and orbit*; jugal reaching the ventral end of a circular antorbital fenestra; subtrapezoidal external supratemporal fenestra, occupying close to 70% of skull roof width; absence of anterior floor of supratemporal fossa; flat dorsal surface of parietal in occipital view; distinct step on dorsal surface of the posterolateral process of squamosal; broad palatine anterior margin, forming a bread transversal suture with maxilla; paired parasagittal ridges on the ventral surface of the basisphenoid; convex dorsal edge of surangular and craniomandibular articulation located above the level of dentary toothrow; ziphodont teeth; humerus with vertical orientation of insertion area of M. subscapularis above internal tuberosity; distal end of deltopectoral crest curves medially surpassing the lateromedial midpoint of humeral shaft; anterior surface of distal end of humerus separated from shaft by a distinct step forming a shelf.

### Description and comparisons

**Skull and jaws.**  In dorsal view the skull is trapezoidal-shaped due to the abrupt end of the rostrum (Fig 2). The supratemporal openings are visible only in dorsal view, and are located on the posterior half of the skull table. The lower jaw is robust and as deep as the skull at the level of the postorbital bar, indicating an extensive area for muscular attachment in the adductor chamber. The maximum length of skull is 49 cm taken from tip of the snout to the tip of mandibular retroarticular process (about 20% longer than in *Uberabasuchus* [CPPLIP 630] and *Hamadasuchus* [ROM 52620]). The maximum transverse width of the skull is at the level of the quadratojugals and the maximum height of articulated skull and jaws is located at the level of the anterior end of the postorbital and the anterior end of the external mandibular fenestra.

In lateral view the dorsal profile of the skull of *Kostensuchus* gen. nov. is almost straight (Fig 2) due to the strong dorsoventral height of the rostrum, as in *Gasparinisuchus* and *Hamadasuchus*, but differing from other peirosaurids (e.g., *Uberabasuchus*, *Montealtosuchus*) [33,34] in which the snout dorsoventral height tapers rapidly towards the anterior end. Sutures among some cranial (e.g., frontal, prefrontal, parietal) bones are difficult to discern due to the extensive pitted ornamentation that covers the external surface of the skull, a condition that is indicative of a fully grown ontogenetic stage (in agreement with the fused condition of the neural arches and centra of cervical vertebrae) [35].

The external nasal apertures are elliptical and anterolaterally oriented, whereas the orbits face laterally (Fig 3). The antorbital fenestra is rounded and small, being approximately 25% the size of the orbits and is dorsoventrally aligned with the ventral margin of the orbit. The ventral margin of the snout is festooned, and teeth are restricted to the rostral half of the skull (ending anteriorly to the level of the orbit). The infratemporal fenestra is subtriangular, anteroposteriorly longer than dorsoventrally high, and about 66% the length of the obit (Fig 2). This fenestra faces laterodorsally, due to the lateral placement of the infratemporal bar relative to the upper temporal bar.

The rostrum is as dorsoventrally deep as transversely wide (Figs 2 and 3). The lateral surface of the rostrum formed by the premaxilla and maxilla are subvertically oriented. This condition resembles that of *Hamadasuchus*, *Lomasuchus* and *Uberabasuchus*, but differs from *Montealtosuchus* and *Gasparinisuchus* in which the rostrum is dorsoventrally shallower relative to its width. The snout roughly represents slightly over 50% the length of the skull, so that the orbits are located just caudal to the mid-length of the skull. Other peirosaurids, including *Uberabasuchus* and *Montealtosuchus*, have a proportionately longer rostrum that occupies close to 60% of the total skull length. The surface of the snout is profusely ornamented, and only bears a tenuous longitudinal line that may correspond with the suture between right nasal and maxilla but the nasal-nasal suture cannot be traced so we interpret they are fused to each other (Fig 2).

The snout is transversely constricted at the level of the premaxilla-maxilla suture, forming a large and laterally open notch for the reception of the hypertrophied fourth dentary tooth (Fig 2). Anterior to this point the premaxilla has a large lateral bulge, caused by the alveolus of the hypertrophied fourth premaxillary tooth, and then narrows anteriorly to end in a pointed premaxillary process that meet the nasals forming a complete internarial bar (Figs 2 and 4). In dorsal view, the

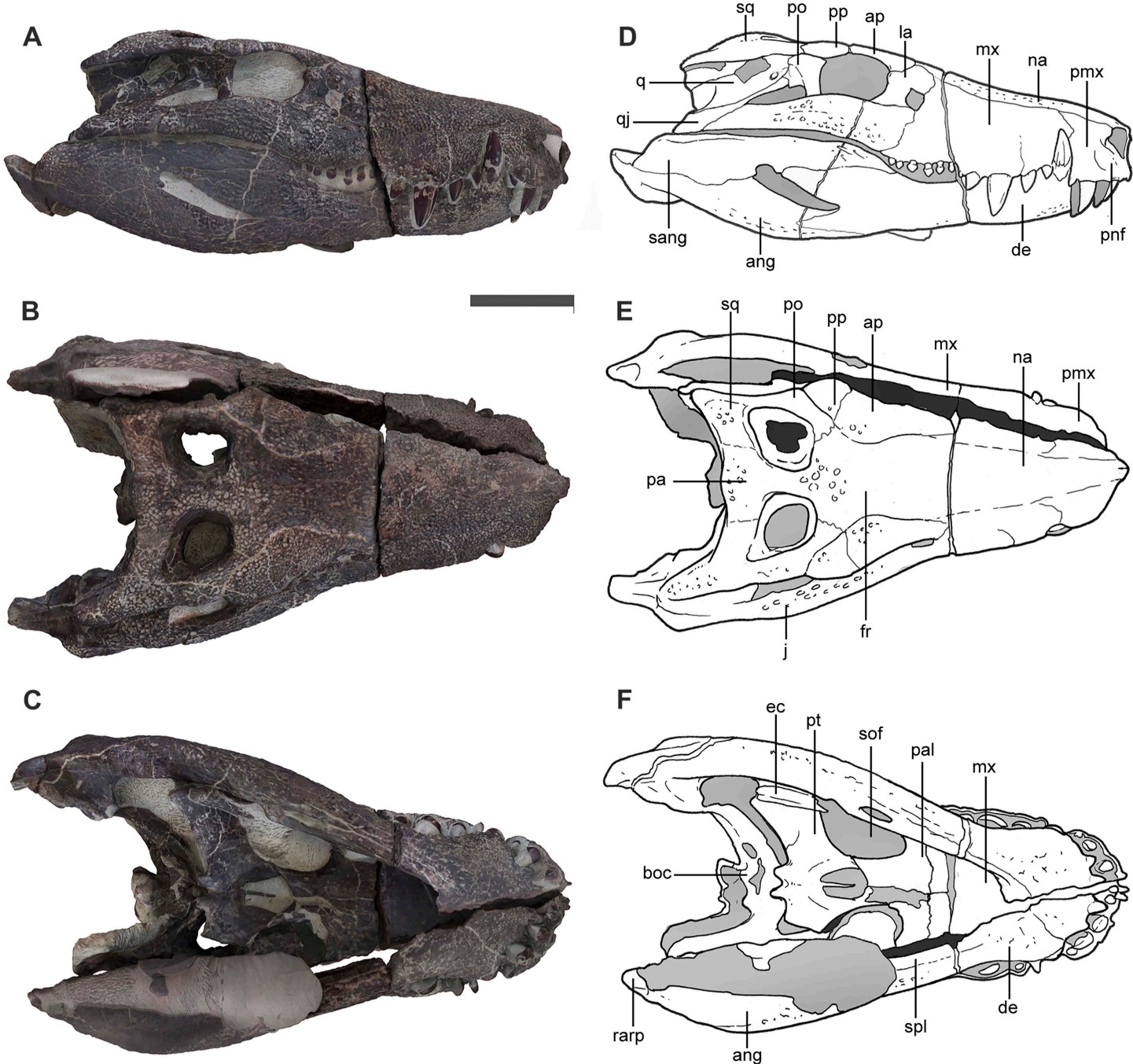

**Fig 2. Skull and jaw of *Kostensuchus atrox* gen. et sp. nov.** Photographs in (A) right lateral, (B) dorsal, and (C) ventral views. Interpretative drawings in (D) right lateral, (E) dorsal, and (F) ventral views. Abbreviations: ang, angular; ap, anterior palpebral; de, dentary; ec, ectopterygoid; fr, frontal; j, jugal; la, lacrimal; mx, maxilla; pa, parietal; pal, palatine; pmx, premaxilla; pnf, perinarial fossa; po, postorbital; pp, posterior palpebral; pt, pterygoid; q, quadrate; qj, quadratojugal; na, nasal; rarp, retroarticular process; sang, surangular; sof, suborbital fossa; spl, splenial; sq, squamosal; stf, subtympanic foramen. Scale bar 5 cm.

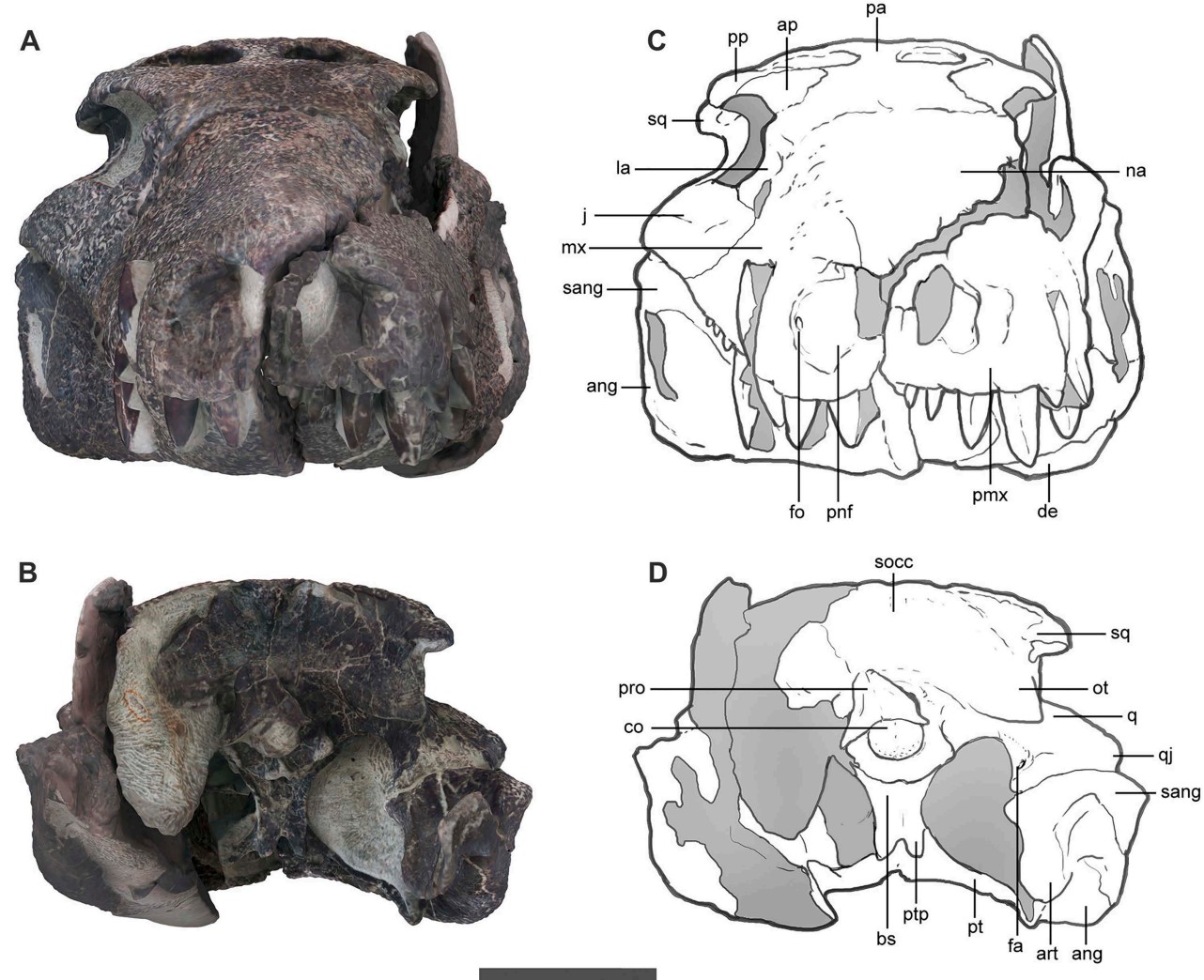

**Fig 3. Skull and jaw of *Kostensuchus atrox* gen. et sp. nov.** Photographs and interpretative drawings in (A-B) anterior and (C-D) posterior views. Abbreviations: ang, angular; ap, anterior palpebral; art, articular; bs, basisphenoid; co, occipital condyle; de, dentary; fa, foramen aërum; fo, perinarial foramen; fr, frontal; j, jugal; la, lacrimal; mx, maxilla; ot, otoccipital; pa, parietal; pmx, premaxilla; pnf, perinarial fossa; p pp, posterior palpebral; pro, proatlas; pt, pterygoid; ptp, posterior pterygoid process; q, quadrate; qj, quadratojugal; na, nasal; rarp, retroarticular process; sang, surangular; socc, supraoccipital; sq, squamosal. Scale bar 5 cm.

sharply pointed internarial bar contrasts with the transversely wide and curved arcade of the rostral margin of the paired premaxillae. The premaxillary component of the internarial bar is projected anterolaterally forming an angle of approximately 60 degrees with the alveolar margin and surpasses anteriorly the level of the first premaxillary tooth (Fig 4B). A large and complete internarial bar is present in *Uberabasuchus* but their presence in other peirosaurids cannot be confirmed due to incompleteness of this delicate part of the skull (e.g., *Montealtosuchus*, *Peirosaurus, Lomasuchus*). This septum, however, is certainly absent in most other notosuchian clades, including baurusuchids (*Baurusuchus, Stratiotosuchus*) and sphagesaurians (*Caipirasuchus, Sphagesaurus*). *Kostensuchus* gen. nov. and *Uberabasuchus* share a similarly rostrally arched internarial bar, although in *Uberabasuchus* the bar is more anteriorly projected than in *Kostensuchus* gen nov.

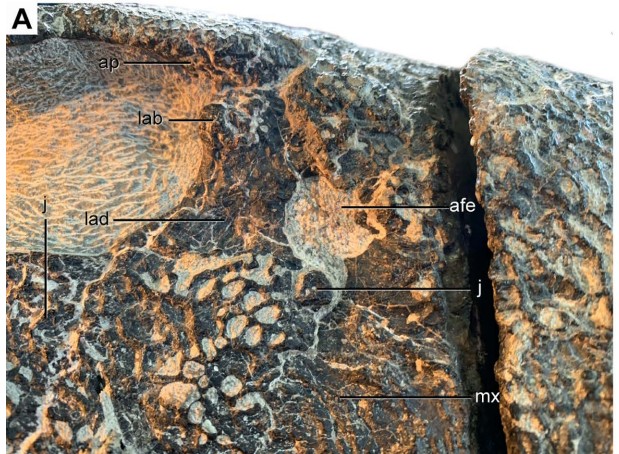
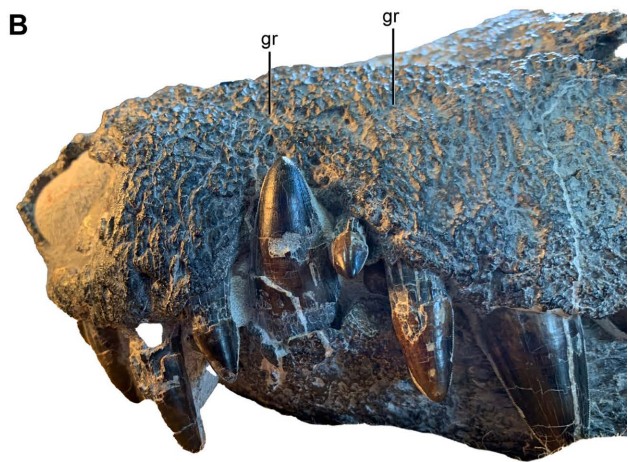

**Fig 4. Details of rostral anatomy of *Kostensuchus atrox* gen. et sp. nov.** Photographs of right antorbital region in lateral view (A) and left surface of rostrum in anterolateral view (B). Abbreviations: afe, antorbital fenestra; ap, anterior palpebral; lab, lacrimal bulge; j, jugal; lad, lacrimal depressed surface; gr, longitudinal groove of maxilla; j, jugal; mx, maxilla.

The posterior and ventral margins of the narial openings are surrounded by a broad and smooth perinarial fossa that extends ventrally reaching the alveolar margin along the level of the first two premaxillary teeth (Figs 2 and 3). The presence of a perinarial fossa is a widespread feature among notosuchians [26], but the large fossa reaching the alveolar margin of *Kostensuchus* gen. nov. resembles in size and extension only that of baurusuchids and peirosaurids, such as the one present in *Uberabasuchus*, *Hamadasuchus*, and *Peirosaurus*. At the posterodorsal corner of the narial opening the premaxilla bears a notch and a small shelf that overhangs the posterodorsal region of the perinarial fossa. The right perinarial fossa bears a single and large neurovascular foramen (Fig 3), present also in *Uberabasuchus* and *Hamadasuchus.* The presence of this foramen cannot be determined with certainty on the left side due to partial damage.

The maxilla of *Kostensuchus* gen. nov. has a festooned alveolar margin, ventrally extended at the level of the third maxillary tooth, as in other peirosaurids, and a smaller posterior ventral outgrowth at the level of the antorbital fenestra (Fig 2). The latter is also present in *Hamadasuchus* but absent or much smaller in other peirosaurids (e.g., *Uberabasuchus*, *Lomasuchus*, *Montealtosuchus*). In *Kostensuchus* gen. nov. the lateral surface of maxilla exhibits (on both sides) a longitudinal and sigmoid groove, extending backwards from the posterior margin of the nasal aperture (Fig 4). It runs approximately parallel to the alveolar margin of maxilla, describing a ventral curvature behind the notch for the mandibular caniniform (Fig 4; S1 File). This groove separates two different decoration patterns on the lateral surface of the snout: above the groove predominate large pits closely positioned to each other, and below the groove the pits are much smaller. *Uberabasuchus* and *Montealtosuchus* also exhibit such a longitudinal and sigmoid groove, although it is located more ventrally than in *Kostensuchus* and does not seem to limit different types of surface ornamentation. *Lomasuchus* clearly has two different types of ornamentation on the maxillary surface, but the groove is much more subtle than in *Kostensuchus* gen. nov. and *Uberabasuchus*. It is not clear the anatomical significance of such a groove, but it approximately follows the course of the dorsal end of the alveoli and the course of the paired maxillary artery and vein that runs within the maxilla [36]. Although with some variation, this feature seems to be so far exclusive of South American peirosaurids as it is absent in *Hamadasuchus* and mahajangasuchids. The maxilla forms the entire anterior margin of the antorbital fenestra and lacks a well-developed antorbital fossa, although a small surface at the anteroventral corner of the antorbital fenestra is smooth (Fig 4A). The nasals of *Kostensuchus* gen. nov. are heavily ornamented and their sutural contacts cannot be clearly determined except for their suture with the premaxilla and maxilla at the anterior region of the rostrum (Fig 2).

The lacrimal external surface is laminar and the central part of its lateral surface bears a fainter ornamentation than the rest of the rostrum between the orbit and the antorbital fenestra (Fig 4A). This faintly ornamented surface is slightly depressed relative to the dorsal and ventral ends of the lacrimal. This morphology differs from that of other South American peirosaurids (e.g., *Uberabasuchus*, *Montealtosuchus*) and uruguaysuchids, which have a well-developed vertical ridge on the lacrimal that separates a smooth and deeply depressed antorbital fossa from the posteriorly located ornamented surface of the lacrimal. It also differs from the condition of *Hamadasuchus* that lacks a lacrimal antorbital fossa and has its entire lateral surface as ornamented as the rest of the rostrum. The dorsal end of the lacrimal bears an ornamented bulge located close to the orbital margin, at its contact with the prefrontal and the anterior palpebral (Fig 4A). A similar but less developed bulge is present in *Montealtosuchus* and *Uberabasuchus* but not in *Hamadasuchus*.

The jugal external surface is flat, dorsoventrally high below to orbit, and heavily ornamented as in other peirosaurids. The jugal extends anteriorly to the orbit and reaches the posteroventral corner of the antorbital fenestra (Figs 2 and 4A), resembling the condition of other peirosaurids (*Uberabasuchus*, *Montealtosuchus*, *Hamadasuchus*). The jugal dorsoventral depth decreases gradually posterior to the orbit and has a long overlapping contact with the quadratojugal along which it tapers to a pointed end located close to the craniomandibular articulation. The ascending process of the jugal is superficial at its anterior end, rather than inset medially as in neosuchians. The posterior margin of the ascending process, however, in inset relative to the lateral surface of the jugal, a condition also present in other peirosaurids (*Uberabasuchus*, *Montealtosuchus*, *Hamadasuchus*), *Kaprosuchus*, and some uruguaysuchids (*A. patagonicus*, *A. wegeneri*). This process is robust and cylindrical and laterally overlaps the descending process of the postorbital (Fig 2).

The sutural contacts of the prefrontal with other elements of the skull are difficult to discern but its posterior contact with the frontal seems to be located at the anteroposterior midpoint of the orbit. From this point forward the prefrontal constraints the width of the frontal and reaches the posterior end of the nasal. The lateral margin of the prefrontal is completely bounded by a large and triangular shaped anterior palpebral that overhangs the orbital opening, covering its anterodorsal region (Fig 2). The anterior palpebral is broadly sutured to the frontal and to the posterior palpebral. The posterior palpebral is shorter than the anterior palpebral and is also triangular shaped but with its base facing anteriorly and its pointed apex directed posteriorly. Its anterior flat margin is strongly sutured to the anterior palpebral and its medial margin is sutured to the postorbital. The posterolateral end of the posterior palpebral overhangs the orbit at the level of the descending process of the postorbital. This configuration of the palpebrals resembles that of other peirosaurids (*Montealtosuchus*, *Lomasuchus*, *Uberabasuchus*).

The frontal lateral margin is sutured to the anterior palpebral so it is does not participate from the lateral margin of the orbit. The dorsal surface of the frontal is triangular shaped, flat, and heavily ornamented and lacks any signs of a sagittal crest or elevated orbital rims (Fig 2). The interorbital region of the frontal is broad relative to its posterior end, which forms part of the anteromedial margin of the external supratemporal fenestra, as in other peirosaurids (*Lomasuchus*, *Uberabasuchus*, *Montealtosuchus*, *Rukwasuchus*) but not in *Hamadasuchus*.

In *Kostensuchus* the temporal region is anteroposteriorly longer relative to the skull length than in other peirosaurids, such as *Uberabasuchus* or *Lomasuchus* that have a larger rostrum and a posteriorly displaced temporal region. The external supratemporal fenestra of *Kostensuchus* gen. nov. are broader posteriorly than anteriorly as in other peirosaurids. The supratemporal fossa bears a large posterior floor formed by the parietal and squamosal but lacks a development of a frontal-postorbital floor at the anterior end (Fig 2), differing from the anterior floor of the supratemporal fossa in other peirosaurids (*Montealtosuchus*, *Uberabasuchus*, *Lomasuchus*, *Hamadasuchus*). The upper temporal bar (formed by the squamosal and postorbital) is relatively narrow (Fig 2B). The width of this surface is approximately 30% the lateromedial width of the external supratemporal fenestra, as in *Uberabasuchus* and *Hamadasuchus*, differing from the broader condition of the postorbital-squamosal bar in *Montealtosuchus* and *Lomasuchus*. The parietal is a single element, as in other crocodylomorphs [37], and its dorsal surface is flat and deeply ornamented. The parietal width between the supratemporal fenestra is approximately 30% the width of each of these openings (Fig 2), resembling the condition of *Lomasuchus* but

differing from both the proportionately wider parietal bar of *Uberabasuchus* and *Montealtosuchus* and the much narrower parietal bar of baurusuchids. The parietal dorsal surface is flat in occipital view as in *Gasparinisuchus* and *Hamadasuchus*, lacking the distinct central concavity present in other peirosaurids (*Montealtosuchus*, *Uberabasuchus*, *Lomasuchus*, *Rukwasuchus*).

The postorbital has a narrow exposure on the skull roof and has an oblique anterolateral margin (Fig 2), as in all notosuchians [26]. This margin is sutured to the posterior palpebral and the postorbital-squamosal contact is an oblique suture located at the anteroposterior midpoint of the supratemporal fenestra. The descending process of the postorbital is inset relative to the skull roof, as in all mesoeucrocodylians (e.g., *Notosuchus*, *Araripesuchus*, *Goniopholis*, *Caiman*). The lateral surface of the descending process is deeply concave because the otic recess extends up to the anterior end of the postorbital (Fig 2), as in other notosuchians [26]. This concave smooth surface faces posterolaterally and contacts the quadrate and quadratojugal along its posterior margin, and the ascending process of the jugal at its anteroventral end. The squamosal forms the posterolateral corner of the skull roof and has a long posterolateral process that reaches the anteroposterior level of the craniomandibular articulation (Fig 2). The length of the posterolateral process resembles the condition of some peirosaurids (*Lomasuchus*, *Uberabasuchus*, *Montealtosuchus*) but is longer than in *Hamadasuchus* and *Rukwasuchus*. The dorsal surface of the posterolateral process is ornamented and bears a subtle step located at its midpoint, so that the posterior region of the posterolateral process is ventrally recessed relative to the dorsal surface of the skull roof, differing from other peirosaurids that lack this step. The posterolateral process of the squamosal is straight and slightly ventrally deflected (as in *Hamadasuchus*), but not as much as in *Lomasuchus*, *Uberabasuchus*, and *Montealtosuchus*. The condition of all peirosaurids however differs from the condition of baurusuchids (e.g., *Baurusuchus*, *Pissarrachampsa*) in which this process is strongly arched and ventrally projected, almost reaching the same level as the quadrate condyles. Within the otic recess the squamosal forms the dorsal half of the otic aperture and has an extensive sutural contact with the quadrate posterior to the otic aperture (Fig 2). The otic aperture is subrectangular with an angled corner at its anterior margin, as in other peirosaurids. The quadrate forms the ventral half of the otic aperture and bears a small foramen just anterior to the otic aperture (Fig 2), referred as the subtympanic foramen [*sensu* 38]. This is the only accessory foramen of the otic recess, resembling the condition of other peirosaurids, sebecids, and uruguaysuchids, but differing from the elliptical openings present in some notosuchians (e.g., *Notosuchus*), baurusuchids, and more basal crocodyliforms (e.g., *Protosuchus, Burkesuchus*). The articular region of the quadrate is relatively short and broad and forms together with the quadratojugal a robust craniomandibular articulation (Fig 3). A small siphoneal foramen is present on the posterior surface of the quadrate, just above the medial articular condyle. The quadratojugal has a large ornamented posteroventral region and a narrower and smooth ascending process that forms the posterior margin of the infratemporal fenestra (Fig 2). The anterior process of the quadratojugal is absent so that this bone fails to form the ventral margin of the infratemporal fenestra, as in other peirosaurids.

The parietals and squamosals form a continuous occipital ridge that projects posteriorly as a slightly developed nuchal crest (Fig 3). This crest is slightly dorsally convex in occipital view as the lateral region of the squamosal is slightly deflected ventrally. The occipital surface formed by the squamosal and otoccipitals is subrectangular in shape and almost as deep as wide (Fig 3).

The palatines form the posterior region of the secondary palate and their suture with the maxillae is transversely oriented and slightly invaginated at its midpoint. The palatines form most of the anterior margin of the suborbital fenestra (Fig 2). The palatine region between the suborbital fenestrae is anteroposteriorly short and lateromedially broad. The posterior edge of the palatines forms a broad V-shaped anterior margin of the choana. The pterygoids are fused into a single element as in all crocodyliforms. The pterygoid bears a deep central depression along its anteromedial region that forms the choanal opening. The choanal groove broadens anteriorly and reaches its maximum width towards the pterygoid-palatine contact. A thin and low choanal septum extends within the choanal groove (Fig 2). The pterygoid flanges are tabular shaped, directed posterolaterally, and its anteroposterior length is approximately 60% of its lateromedial extent. The

pterygoid bears two noticeable posterior processes with a narrow U-shaped notch between them, located at its postero-medial region (Fig 2). Only the posterior end of the right ectopterygoid is exposed, which is long and narrow and borders the pterygoid flange.

The lower jaw of *Kostensuchus* is dorsoventrally deep, especially at its posterior end which is slightly deeper than the posterior half of the skull (Fig 2). The jaw is stouter than in other peirosaurids. The symphyseal region is transversely broad in *Kostensuchus* (Fig 5), being its width/length ratio approximately 65% (including the splenial participation in the symphysis). This represents the broadest mandibular symphysis among all peirosaurids, in which broad snouted forms have width/length ratios close to 60% (e.g., *Colhuehuapisuchus*, *Gasparinisuchus*, *Barrosasuchus*, *Miadanasuchus*) and narrow snouted peirosaurids have ratios ranging between 40% and 50% (*Montealtosuchus*, *Uberabasuchus*, *Patago-suchus*, *Bayomesasuchus*). An event narrower symphysis is present in *Hamadasuchus* (ranging 30%) and certainly in the long snouted itasuchids (*Itasuchus*, *Pepesuchus*). As in other broad snouted peirosaurids, the rostral margin of the dentaries anterior the caniniform meet at a broad angle (approximately 140 degrees). The dentary is bulged at the level of the caniniform and constricted posteriorly to it, forming a lateral concavity where the first enlarged maxillary caniniform occludes (Fig 5). This constriction is present in most peirosaurids, except for *Colhuehuapisuchus* in which the dentaries are almost straight posterior to the lower hypertrophied tooth (which in turn would indicate a less developed maxillary caniniform than in *Kostensuchus* gen. nov.).

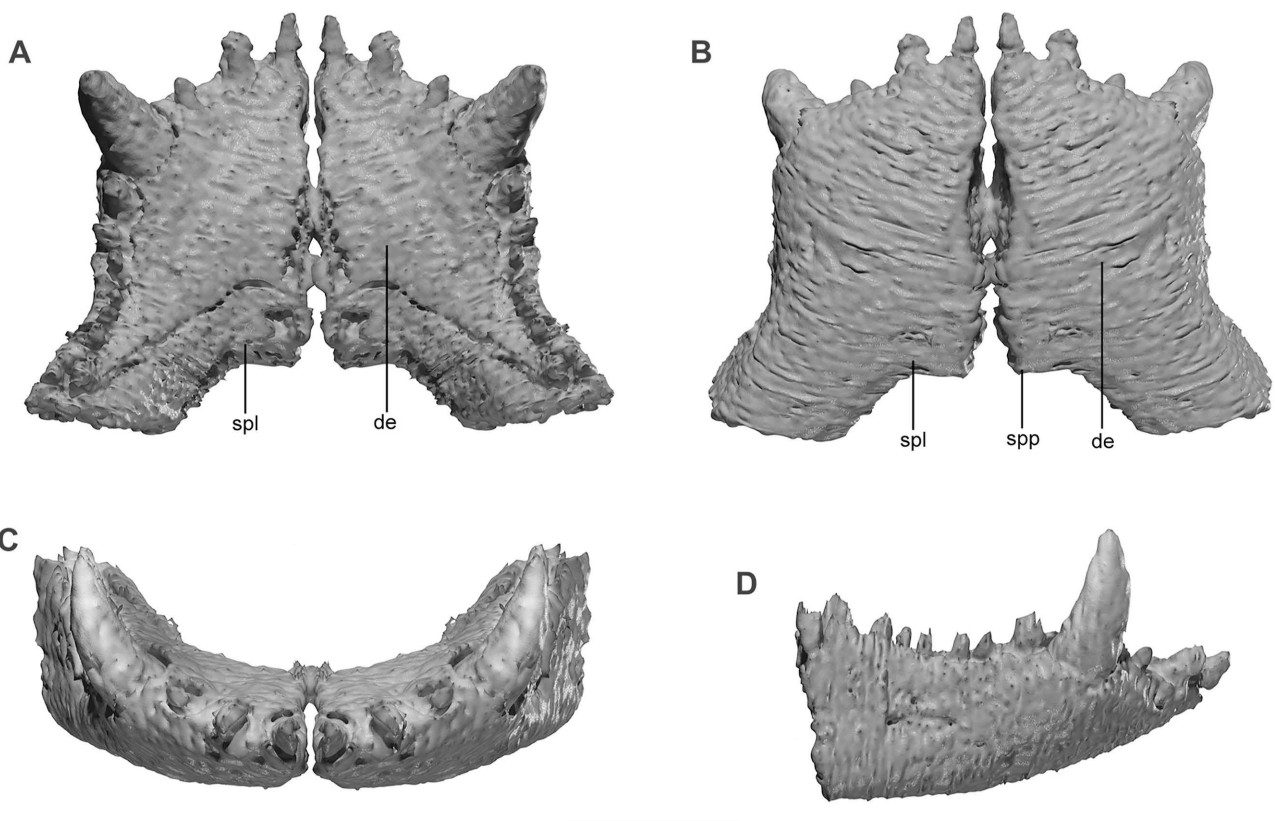

**Fig 5. Virtual threedimensional model of anterior region of lower jaw of *Kostensuchus atrox* gen. et sp. nov.** Model in (A) dorsal, (B) ventral, (C) anterior, and (D) right lateral views. Abbreviations: de, dentary; spl, spenial; spp, splenial peg. Scale bar 5 cm.

The dentary Is the longest bone of the lower jaw, representing twice the length of the angular plus retroarticular (Fig 2). In lateral view the dentary tapers gradually towards its anterior end as in other peirosaurids, uruguaysuchids, and sphagesaurians, but differs from the strongly angled profile of baurusuchids and some sebecoids (e.g., *Iberosuchus*, *Bretesuchus*). The lateral surface shows a change in ornamentation pattern: along the anterior half, the surface is ornamented with elliptical pits of small size whereas the posterior half is gradually shifts to an ornamentation composed by a complex pattern of oblique ridges and grooves (close to the external mandibular fenestra; Fig 2). The posterodorsal region of the dentary bears a broad longitudinal sulcus below the buccal margin, a feature shared with *Uberabasuchus* and *Montealtosuchus*. The splenial forms part of the mandibular symphysis, but this contribution in ventral view is less than 20% of the symphyseal length (Fig 5). In both ventral and dorsal views, the splenial-dentary suture is transversely oriented, as in the broad snouted *Colhuehuapisuchus* and *Miadanasuchus*, rather than V-shaped as in other peirosaurids. The posterior end of the splenial-splenial suture bears a posterior peg, a common feature with several peirosaurids and other notosuchians. The splenial bears a notably large slot-like foramen intermandibularis oralis posterior to the symphyseal region, a feature that is also present in the broad snouted *Colhuehuapisuchus* and *Miadanasuchus.* The splenial is broadly exposed on the ventral surface of the mandibular ramus along its contact with the dentary and angular (Fig 2). The medial surface of the splenial is only exposed along its posterior region, which is flat, smooth, and completely sutured with the angular, denoting the absence of a foramen intermandibularis caudalis as in all notosuchians.

The angular and surangular are also ornamented along their lateral surface (Fig 2), with pits being more frequent at its posterior end and ridges more frequent anteriorly (especially in the surangular). The angular is sutured to the dentary by an oblique suture that runs posteroventrally from the acute anterior end of the external mandibular fenestra. The dorsal margin of the angular forms the complete ventral margin of the external mandibular fenestra. The angular-surangular suture runs posteriorly and slightly ventrally from the posterior end of the mandibular fenestra (Fig 2), so that most of the lateral surface of the mandibular ramus is formed by the surangular at the level of the craniomandibular articulation. The ventral margin of the angular is slightly curved dorsally towards its posterior end and bears a very low and broad angular ridge at the level of the external mandibular fenestra. This contrasts with the sharper and more prominent angular ridge of *Uberabasuchu*s and *Montealtosuchus* that extends posteriorly surpassing the level of the craniomandibular articulation.

The anterior end of the surangular is forked at its anterior contact with the dentary. The ventral margin of the surangular has a lineal overlapping suture with the dentary and this contact reaches the posterodorsal corner of the external mandibular fenestra (Fig 2). The surangular forms the rounded posterior edge of this opening and is sutured to the angular at the posteroventral corner of this fenestra. The posterior region of the surangular bears a broad longitudinal ridge that extends posteriorly to the level of the glenoid facet and borders the anterior region of the retroarticular process (Fig 2), as in other peirosaurids (e.g., *Uberabasuchus*, *Montealtosuchus*). The surangular forms most of the lateral flange of the retroarticular process. The lateral flange is triangular, dorsally concave, and is separated from the surangular contribution to the glenoid facet by an elevated transversal ridge. The articular forms part of the lateral flange of the retroarticular process and bears a broad and low longitudinal ridge that separates the medial from the lateral flange. This ridge ends posteriorly in a rounded bulge that is directed posterodorsally (as in *Montealtosuchus* but not as dorsally recurved as in *Uberabasuchus*). The medial flange of the retroarticular process extends ventromedially and its slightly concave surface faces posteromedially (Fig 3). The ventral margin of this flange is convex as other peirosaurids and uruguaysuchids, but not as extended and paddle shaped as in sphagesaurians and baurusuchids. The medial flange bears the foramen aereum and a small bulge located close to it (as in uruguaysuchids and neosuchians). The articular forms the medial half of the glenoid facet for the quadrate condyles, which faces dorsally and is separated from the retroarticular process by an elevated transversal ridge.

**Dentition.** *Kostensuchus* gen. nov. exhibits a morphological pattern of the tooth row that resembles that of other peirosaurids, consisting in conical teeth anterior to the large mandibular caniniform, and post-caniniform teeth that have lanceolate shaped crowns and are smaller in size (Fig 2). The upper toothrow has enlarged caniniforms in the

premaxilla and the maxillary and bears a tooth count of 15 elements, which is smaller than that of other peirosaurids (e.g., *Hamadasuchus, Lomasuchus, Montealtosuchus*). All preserved teeth bear small symmetrical denticles along the entire mesial and distal carinae, characteristic of a fully ziphodont dentition.

The premaxilla bears five teeth as in most peirosaurids, the two anteriormost of which are small and closely positioned to each other. The fourth tooth is the largest, followed by the third that is 70% and the fifth one is approximately 50% the size of the largest element (considering the basal mesiodistal width of the crown). The upper toothrow has a long gap between the premaxillary and maxillary dentition that accommodates a large dentary caniniform. The right maxilla has ten teeth exposed on the lateral surface. The first three increase progressively their size up to the third maxillary tooth, which is the largest tooth of the upper row and similar in size to the lower caniniform. The following five maxillary teeth are notably smaller. The posteriormost teeth are short-crowned, lanceolate, and much more buccolingually compressed than the preceding conical-shaped teeth. The change in shape and relative sizes of the teeth in *Kostensuchus* gen. nov. closely resemble that of *Uberabasuchus and Montealtosuchus,* but differs from *Hamadasuchus* in the absence of a second enlarged maxillary in its posterior region. All peirosaurids nonetheless fall within the generalized condition of bearing a large tooth count, compared with baurusuchids or sphagesaurids that have a maxillary tooth count limited to five or six teeth.

The dentary teeth are less well exposed in *Kostensuchus* gen. nov. but CT data, as well as the natural fracture present along the symphyseal region allows determining the procumbent nature of the two anteriormost dentary teeth (Fig 3), as in *Colhuehuapisuchus* but differing from other peirosaurids (e.g., *Montealtosuchus, Uberabasuchus*) in which these teeth are not directed as anteriorly as in these two Patagonian taxa. The fourth dentary tooth is more than twice the length and height of other lower teeth and strongly protrudes dorsally surpassing the dorsoventral midpoint of the snout. The post-caniniform mandibular teeth are not exposed but CT data shows they have a posterior wave of minor size variation.

**Postcranial skeleton.** The postcranial skeleton of *Kostensuchus* gen. nov. is represented by the presacral and sacral regions of the vertebral column as well as a large part of the dorsal shield of osteoderms, scapular girdle, left humerus and partial remains of the right humerus, ilia, and ischia (Fig 6; S1 File).

The cervical series has relatively low anteriormost cervicals that progressively become anteroposteriorly short and dorsoventrally high (Fig 7). The posterior cervicals have high rod-like neural spines, which are as high as the rest of the vertebrae. This feature is also present in *Uberabasuchus*, *Montealtosuchus*, and is also present in some sphagesaurians (e.g., *Notosuchus*) and baurusuchids. The prezygapophyses are directed anterodorsally and bear a straight anterior margin but the postzygapophyses are occluded by sediment or the overlying osteoderms. The cervical centra are poorly constricted and bear a knob-like hypapophysis. The cervical ribs are anteroposteriorly elongated as in all crocodyliforms and bear a distinct knob at the base of the posterior process, which is much more reduced than the well-developed process described for *A. tsangatsangana* and *Mahajangasuchus* [39].

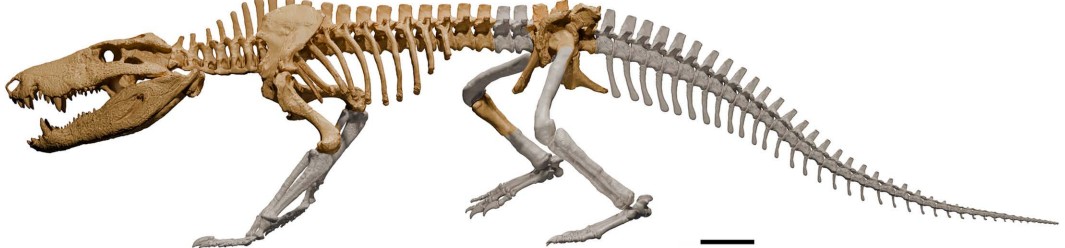

**Fig 6. Skeletal reconstruction of *Kostensuchus atrox* gen. et sp. nov.** Three-dimensional model in left lateral view with preserved bone in light brown and missing elements in light grey. Missing elements were modeled based on selected notosuchians for which these regions are known (e.g., *Montealtosuchus*, *Araripesuchus*), as well as extant crocodylians (*Caiman*). Scale bar 5 cm.

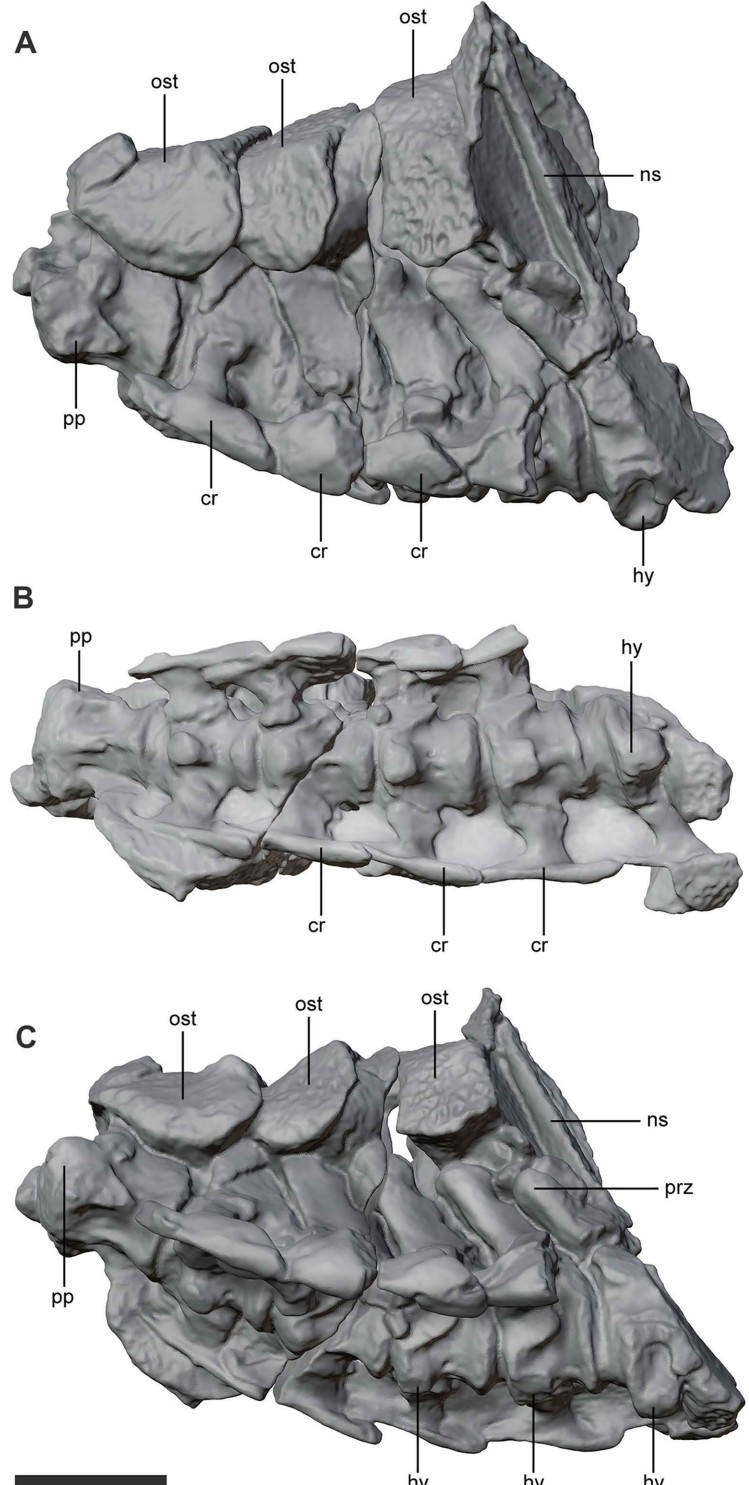

**Fig 7. Cervical vertebrae of *Kostensuchus atrox* gen. et sp. nov.** Threedimensional model of cervicals, from the axis to the seventh cervical vertebrae in (A) left lateral, (B) ventral, and (C) left lateroventral views. Abbreviations: cr, cervical rib; hy, hypapophysis; ns, neural spine; ost, dorsal osteoderm; pp, parapophysis; prz, prezygapophysis. Scale bar 5 cm.

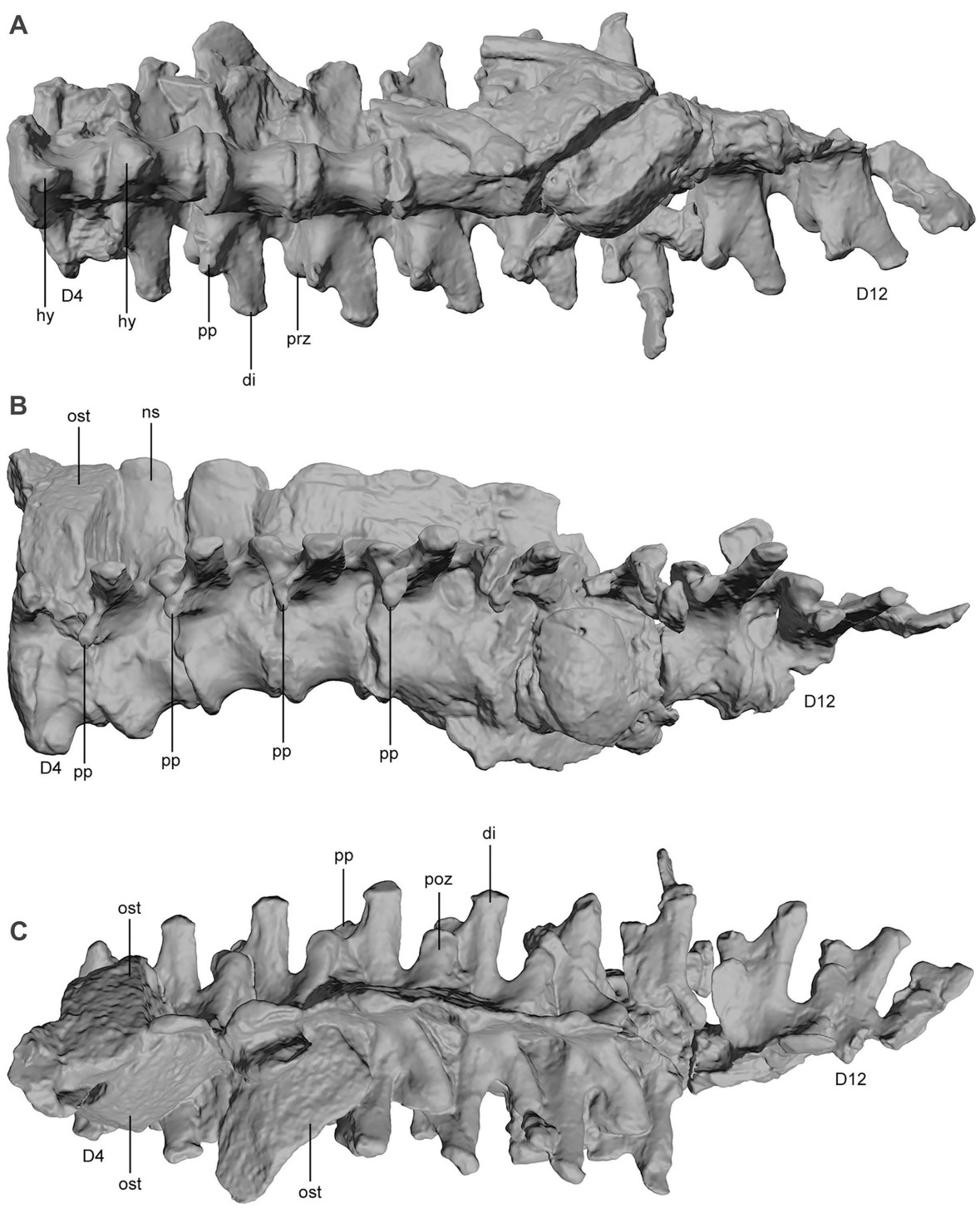

**Fig 8. Dorsal vertebrae of *Kostensuchus atrox* gen. et sp. nov.** Threedimensional model in (A) left ventral, (B) left lateral, and (C) dorsal views of fourth dorsal vertebra (D4) to twelfth dorsal vertebra (D12). Abbreviations: di, diapophysis; hy, hypapophysis; ns, neural spine; ost, dorsal osteoderm; pp, parapophysis; poz, postzygapophysis; prz, prezygapophysis. Scale bar 5 cm.

The dorsals have a much lower neural spine and a proportionately longer and lower centrum and neural arch (Fig 8). The parapophysis migrates gradually along the first five dorsal elements from the dorsal end of the centrum to the same level of the diapophysis. The dorsal neural spines are low and have a convex anterior and concave posterior margin. The posterior region of the dorsal have an enlarged centrum that likely represents a pathology of this specimen (Fig 7B)

The sacrum is composed of two vertebrae, although the absence of a caudal series precludes determining the presence of a caudosacral element, as in *A. gomesii*. The sacrum was preserved in articulation with the last dorsal vertebrae and with the left ilium. The first sacral rib attaches to the ilium at the level of the preacetabular process and is much shorter anteroposteriorly than the second sacral rib. The second sacral rib is wing-shaped and extends posterolaterally exceeding the posterior end of the centrum and neural arch (including the postzygapophyses) of the second sacral vertebra.

The scapulae of *Kostensuchus* gen. nov. are preserved on both sides of the body, close to their original anatomical position (Fig 9). It is dorsoventrally short and anteroposteriorly broad in comparison with other crocodyliforms. In general terms both peirosaurids and uruguaysuchids have a relatively short and broad scapula, but in the case of *Kostensuchus* gen. nov. this feature is even more notable. The dorsal blade of the scapula is approximately 90% of the maximum dorsoventral height of the scapula. The constriction between the ventral and dorsal expansions is limited, being approximately 75% the anteroposterior width of the ventral end of the scapula. The glenoid articular facet of the scapula faces ventrally, rather than posteroventrally as in *Mahajangasuchus*. This region is lateromedially thick, whereas dorsal to the central constriction the scapula becomes laminar (throughout the scapular blade). The coracoid is approximately as long as the scapula and lacks signs of a torsion along its shaft. The coracoid foramen is dorsoventrally elongated and located centrally on the proximal expansion. The glenoid facet faces posterodorsally and its ventral end bears a well-developed lip so that the region just ventral to the glenoid is markedly concave, as occurs in baurusuchids [27].

The humerus (Fig 10) is remarkably robust and distally expanded in comparison with that of other notosuchians and other crocodyliforms. The width of the proximal end is approximately 30% of the humeral length, which does not differ markedly from the range of other taxa. The width of the distal end, however, is 30% wider than the proximal end and

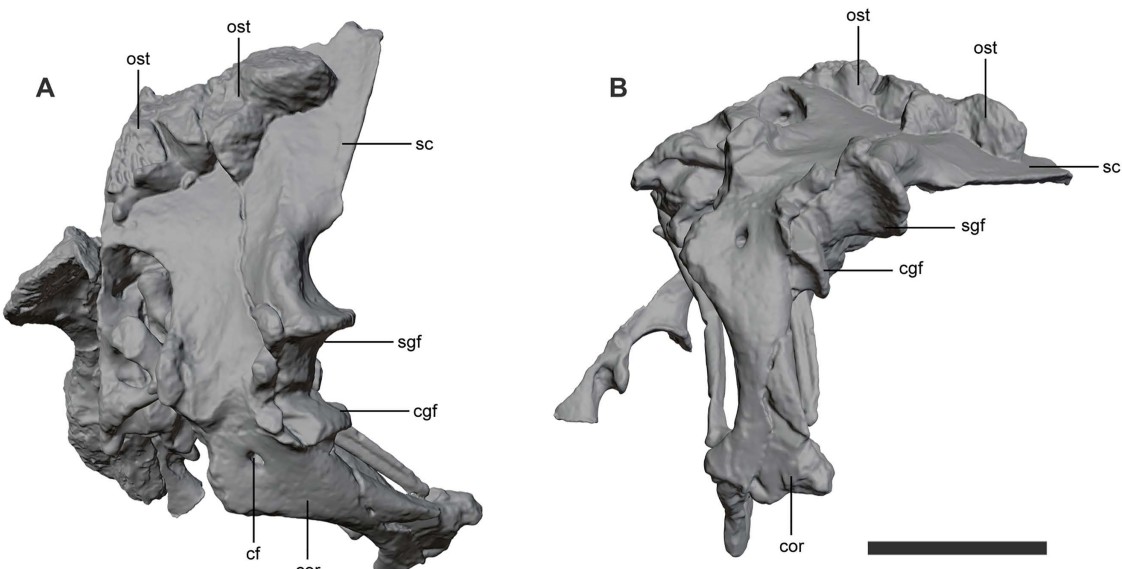

**Fig 9. Shoulder girdle of *Kostensuchus atrox* gen. et sp. nov.** Threedimensional model in (A) left lateral view. Abbreviations: cf, coracoid foramen; cgf, coracoid glenoid facet; cor, coracoid; sc, scapula; sgf, scapular glenoid facet; ost, osteoderm. Scale bar 5 cm.

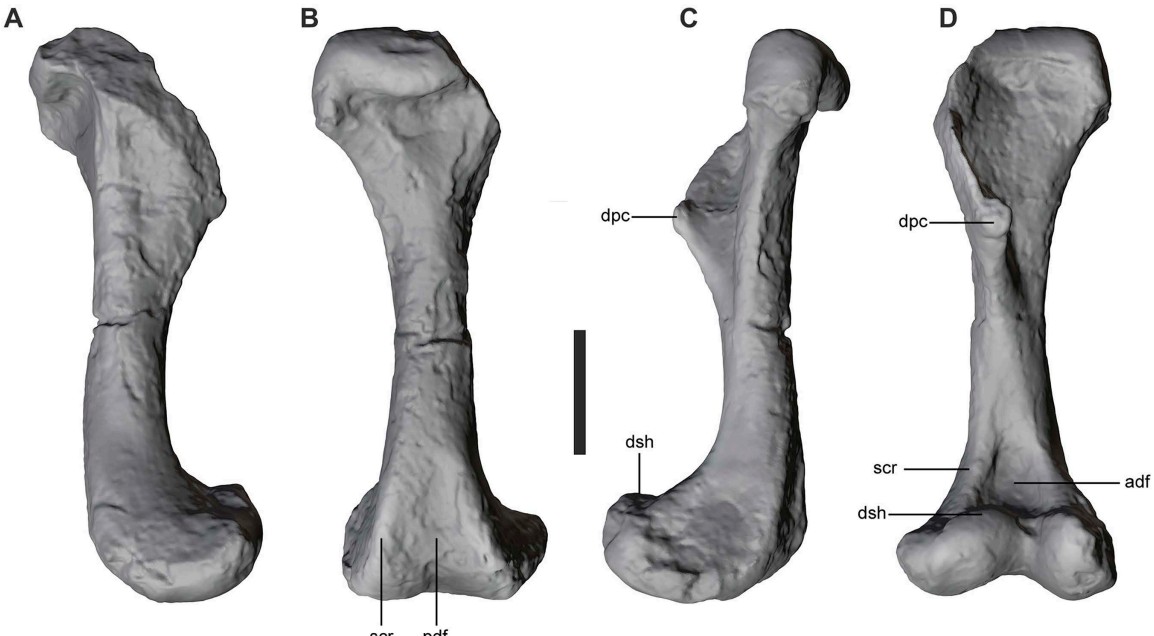

**Fig 10. Humerus of *Kostensuchus atrox* gen. et sp. nov.** Threedimensional model in (A), lateral, (B), posterior, (C), medial, and (D), anterior views. Abbreviations: adf, anterior distal fossa; dpc, deltopectoral crest; dsh, humeral distal shelf; pdf, posterior distal fossa; scr, supracondylar crests. Scale bar 5 cm.

40% of the total humeral length. The proximal end of the deltopectoral crest is damaged but it can be determined that the crest is triangular shaped in lateral view and bears a rounded tubercle at its tip. Distal to the tubercle the crest extends distomedially onto the shaft and reaches or slightly surpasses the lateromedial midpoint of the shaft, as in baurusuchids and sebecids [40]. The expanded distal end bears two prominent condyles and flat lateral and medial surfaces bounded by sharp supracondylar crests. The posterior surface of the humerus bears and a shallow fossa between the posterior supracondylar crests. However, the anterior surface of the distal humerus bears a deep rounded fossa between the supracondylar crests (presenting a morphology not recorded in other crocodyliforms but that may characterize at least some peirosaurids). Distal to this fossa, the anterior surface of the humerus bears a distinct step and a proximally facing shelf that extends lateromedially along the entire width of the distal end of the humerus. This feature has only been noted in *Sebecus icaeorhinus* and *Iberosuchus* but is absent in baurusuchids and in *Mahajangasuchus* [40].

Both ilia of *Kostensuchus* gen. nov. are available: the left one preserved in articulation with the sacrum. The ilium has a well-developed preacetabular process (Fig 11) in comparison with other notosuchians, nonetheless, this process is not as long as in more basal crocodyliforms (e.g., *Protosuchus*). The acetabulum resembles the condition of uruguaysuchids (e.g., *A. gomesii*) in that it is deeper than in neosuchians but lacks the robust and horizontally directed supraacetabular crest that forms a thick acetabular roof in sphagesaurians (e.g., *Notosuchus*), baurusuchids (e.g., *Baurusuchus albertoi*), and to a lesser degree sebecids [40]. The insertion area of the *Mm. iliotibialis* forms a rugose area along the supraacetabular crest at the level of the posterior half of the acetabulum. This scar does not extend anteriorly towards the preacetabular process as in *Sebecus icaeorhinus*. The pubic peduncle is notched, forming deep and narrow incisure with an angle of approximately 30 degrees that resembles that of *Mahajangasuchus* but differs from that of neosuchians or other notosuchians (e.g., *Sebecus*, *Notosuchus*) that have a much broader angle and shallower incisure. The ischial peduncle is broader than the pubic peduncle and bears a flat antitrochanter on its acetabular surface. The postacetabular

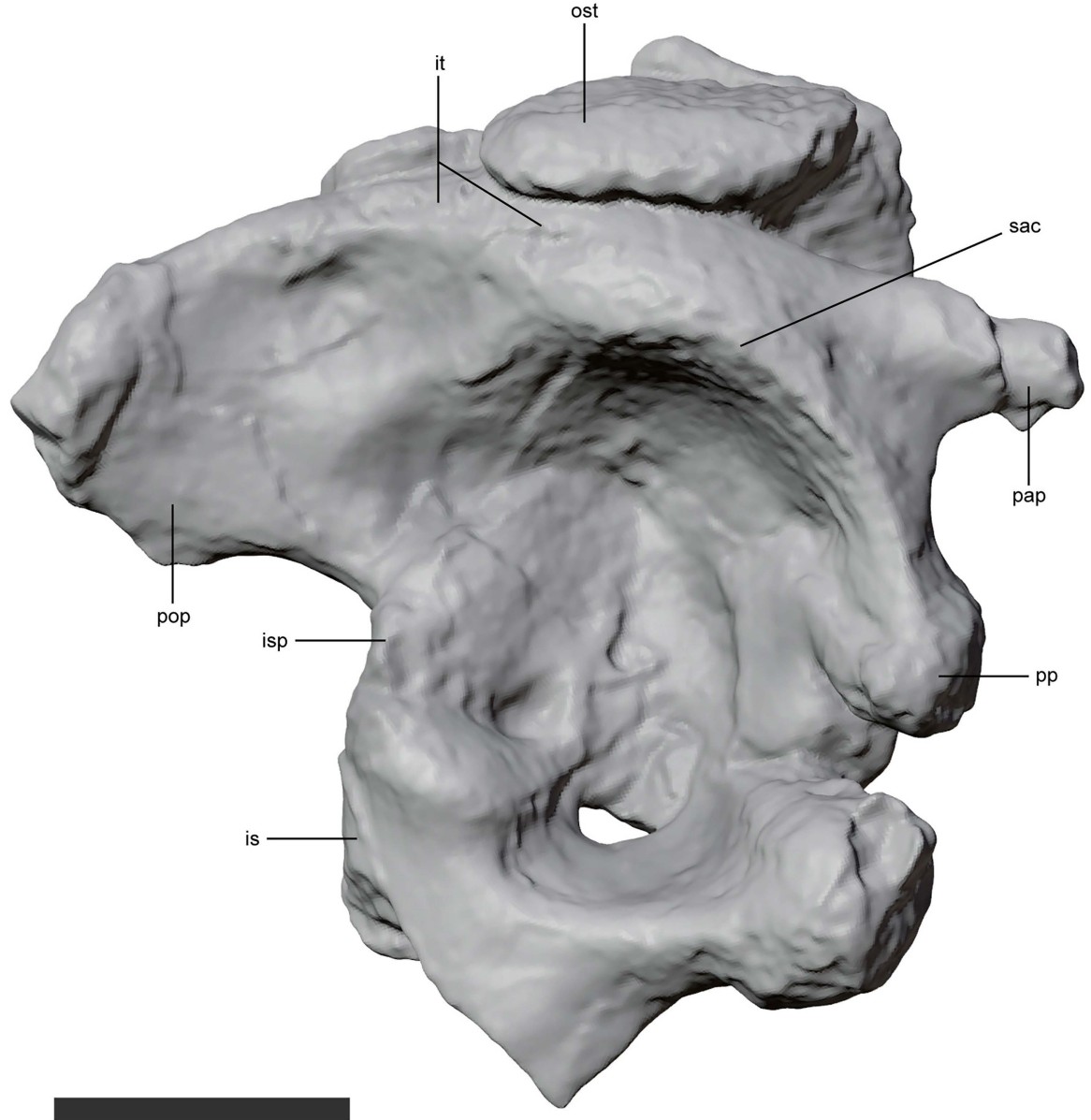

**Fig 11. Ilium of *Kostensuchus atrox* gen. et sp. nov.** Threedimensional model in (A) lateral view. Abbreviations: is, ischium; isp, ischial peduncle; it, insertion area of *M. iliotibialis*; pap, preacetabular process; pop, postacetabular process; pp, pubic peduncle; ost, dorsal osteoderm; sac, supracetabular crest. Scale bar 5 cm.

process is subequal in length to the acetabulum and is dorsoventrally high, being as deep as the acetabulum. The posterior end of the postacetabular process is dorsoventrally deep, with a horizontal ventral margin, and a subrectangular posterior end (as in most notosuchians, except for *Sebecus icaeorhinus*). The proximal end of the ischium contacts the pubic and ischial peduncle of the ilium and forms the ventral end of the acetabulum, excluding the pubis from it, as in all mesoeucrocodylians.

Several osteoderms have been preserved associated to some vertebral segments, such as the cervical and dorsal regions. All vertebral osteoderms are well ornamented with the pitted pattern present in the dermal skull bones, bear a

longitudinal ridge along its length, are subrectangular with a rounded lateral margin, have an overlapping surface along their anterior margin, and lack an anterolateral articular peg. Cervical osteoderms are smaller than dorsal osteoderms, especially in their lateromedial width and have the longitudinal ridge close to their lateromedial midpoint. Dorsal osteoderms, instead, are broader and have the longitudinal ridge located closer to the rounded lateral margin than to the medial margin. There are few isolated osteoderms that are rounded, with their surface pitted, and lack longitudinal ridges, interpreted as accessory dorsal osteoderms.

### Phylogenetic affinities

Peirosaurids are recorded in South America, Africa, and Madagascar and have been recognized as important components of Cretaceous faunas of Gondwana for the last 40 years [41,42]. Currently, there is ample consensus in their monophyly [43] and most phylogenies depict the African *Hamadasuchus* as sister group of South American peirosaurids [25,44,45; but see 46 for an alternative arrangement]. However, at the moment there is a lack of a stable resolution in the internal relationships of Peirosauridae. Several authors have noted the presence of peirosaurids with broad jaws [47–50] but most of them are known from fragmentary mandibular remains. The only exception is *Gasparinisuchus* [47] that is known from a specimen with partially preserved craniomandibular remains (previously referred to *Peirosaurus*) [51]. Although broad-snouted peirosaurids are potentially closely related to each other, so far, these taxa have been rarely included in phylogenetic analyses, likely due to the incompleteness of their remains. The discovery of *Kostensusuchus* with an exquisitely preserved skull allows understanding for the first time this morphological type as well as testing the relationships of broad snouted peirosaurids within the context of other members of this clade.

   Our phylogenetic results (Fig 12; S1 File) support a monophyletic Peirosauridae as in previous analyses [25,26,45] but also retrieves a clade of broad snouted peirosaurids, composed by the Late Cretaceous taxa *Kostensuchus* gen. nov., *Colhuehuapisuchus*, *Miadanasuchus*, and *Gasparinisuchus*. Within this group, *Gasparinisuchus* and *Colhuehuapisuchus* are depicted as sister taxa due to the presence of a straight alveolar margin of the dentary posterior to the lower caniniform (in dorsal view; char. 158−0) rather than the markedly concave margin of other peirosaurids (including *Kostensuchus* gen. nov.). *Kostensuchus* gen. nov. is depicted as the sister taxon of this clade, forming a group that is diagnosed by the presence of procumbent anterior mandibular alveoli (char. 262−1; not preserved in *Gasparinisuchus*). The clade of broad snouted peirosaurids is formed with the addition of *Miadanasuchus oblita* [50,52] from the Maastrichtian of Madagascar. This group is diagnosed by the presence of broad U-shaped symphyseal region of the dentaries (in ventral view; char. 154−1), the presence of a large slot-like foramen intermandibularis oralis facing medially on the splenial (char. 174−1), and a transversely oriented splenial-dentary suture at the mandibular symphysis (on both the ventral and dorsal surface; chars. 185−1, 440−1).

   The clade of South American peirosaurids is retrieved as monophyletic, leaving the African *Hamadasuchus* as the earliest diverging lineage of Peirosauridae. The South American clade is diagnosed by the presence of the frontal participation on the internal supratemporal fenestra preventing the parietal-postorbital contact (char. 23−0, a wedge-like process of the maxilla on the lateral surface of the premaxilla-maxilla suture (char. 213−1), and the palpebrals extensively sutured to each other and to the lateral surface of the frontal (char. 214−1). Peirosauridae is diagnosed by the presence of true ziphodont teeth (char. 120−0) rather than pseudoziphodont as in mahajangasuchids, nasal-maxilla sutures nearly parallel to each other (char. 128−0), basisphenoid with long dorsolateral extension exposed on the lateral surface of the braincase (char. 147−1), large perinarial fossa facing anteriorly toward the alveolar margin (char. 226−2), foramen in perinarial depression (char. 237−1), palatine suture with maxilla transversal and slight invaginated (char. 243−2), and prominent depression on palate near alveolar margin at the level of the sixth or seventh maxillary alveolus (char. 396−1). Further phylogenetic data can be found in S1 File.

   Support values for most nodes are low due to the presence of fragmentary taxa and most high-level nodes within Mesoeucrocodylia are collapsed (see S1 File), but Peirosauridae are among the lower-level clades retrieved with jackknife

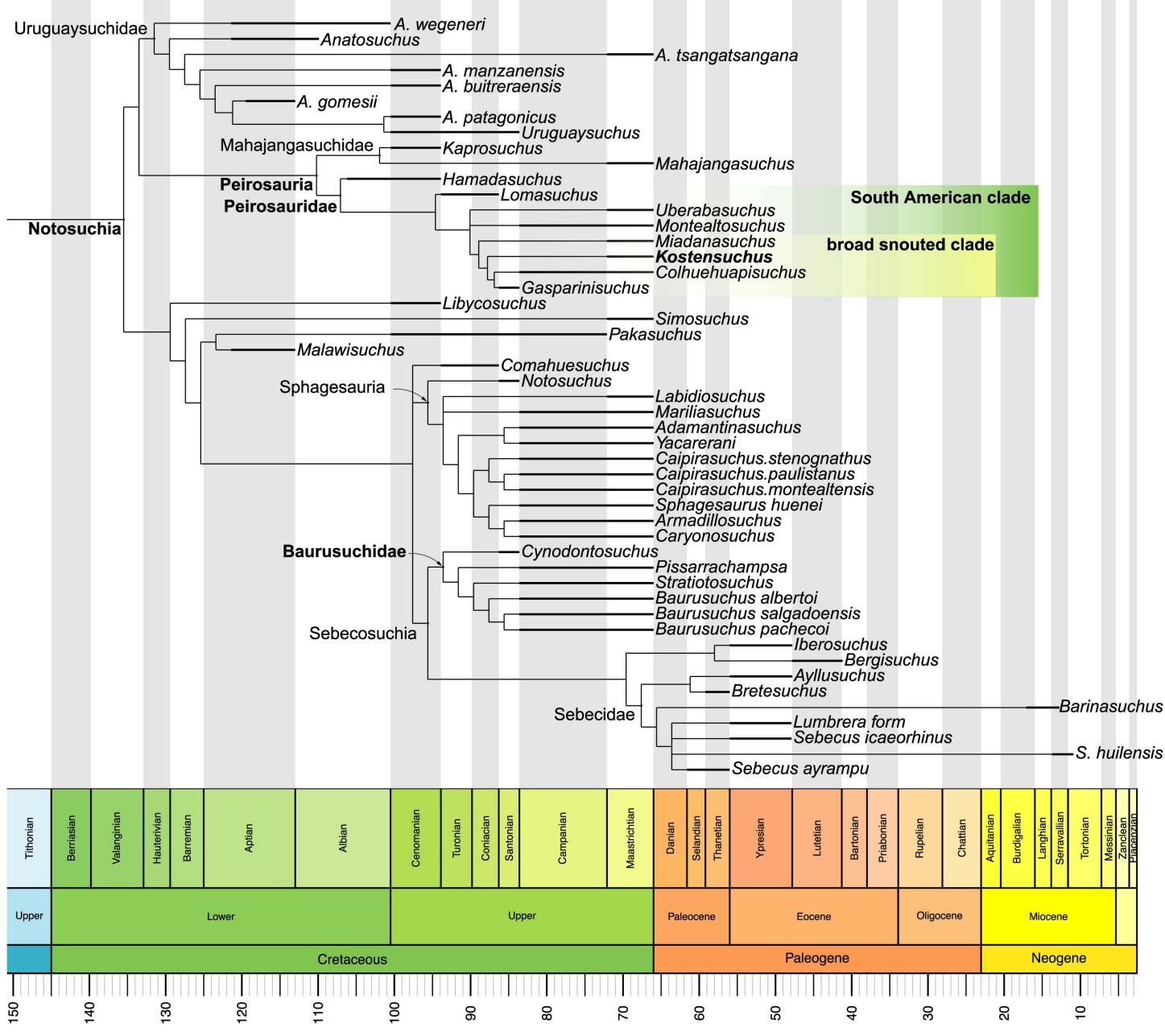

**Fig 12. Phylogenetic relationships of *Kostensuchus atrox* gen. et sp. nov.** Reduced strict consensus tree obtained in the phylogenetic analysis. Thick bars in terminal branches represent the chronostratigraphic uncertainty for each taxon. Phylogenetic tree was calibrated using the R package Paleotree [81].

frequencies higher than 50%, along with Sphagesauria, Mahajangasuchidae, Atoposauridae, Thalattosuchia, Tethysuchia, Goniopholididae, Eusuchia. The impact of fragmentary taxa that became unstable in parsimony jackknife support analysis is evident when the *pcrjak* method [32] is used, which ignores the impact of these wildcards on support values. The *pcrjak* tree retrieves most major nodes within Crocodyliformes with high support (see S1 File). In this tree the support is high also

for Peirosauridae (80%) and for the South American clade of peirosaurids (71%). Furthermore, the close affinity of the broad snouted *Kostensuchus* gen. nov. and *Colhuehuapisuchus* is supported with 79%, given that both *Miadanasuchus* and *Gasparinisichus* are detected as unstable during the jackknife replicates.

## Discussion

### Broad-snouted peirosaurids as apex predators

The discovery of *Kostensuchus* gen. nov. provides the first comprehensive insight into the anatomy of large-bodied, broad-snouted peirosaurids that thrived at the end of the Cretaceous. Represented by a partially complete skeleton (Fig 6), *Kostensuchus* gen. nov. contrasts sharply with the fragmentary remains of other Maastrichtian broad-snouted peirosaurids, such as *Colhuehuapisuchus* and *Miadanasuchus*, which are known only from partial mandibles. *Kostensuchus atrox* gen. et sp. nov. is the largest peirosaurid for which a reliable size estimate can be calculated, with an estimated body length of 3.5 meters and a body mass of 250 kg (see Materials and Methods). This body mass is significantly greater than those of earlier narrow-snouted peirosaurids, whose body mass estimates range between 12 and 63 kg (e.g., *Montealtosuchus*, *Uberabasuchus*, *Lomasuchus*, *Hamadasuchus*) [23]. Thus, peirosaurids appear to align with a broader trend observed in some notosuchian clades, which evolved into larger body sizes over the course of the Cretaceous [53]. This shift has been linked to a transition from omnivorous, medium-sized forms to larger, hypercarnivorous apex predators [54,55].

Broad-snouted peirosaurids were likely top predators, as evidenced by several adaptations indicative of a predatory lifestyle and its large body size. For instance, *Kostensuchus* gen. nov. had a broad, robust snout that comprised slightly over 50% of its total skull length, a large adductor chamber in the skull, and a deep mandibular adductor fossa, suggesting the presence of powerful temporal muscles. This morphology, combined with its exceptionally large conical teeth along the anterior and mid regions of the toothrow, with labiolingually broad teeth with ziphodont margins, results in a lengthening of the shearing edges whose primary functions are puncturing and slicing through the flesh of sizable prey. These features, as well as a reduced tooth count that *Kostensuchus* lacks, have been interpreted as adaptations towards hypercarnivory in baurusuchid crocodyliforms and other large predators capable of subduing large struggling prey [54–57]. This morphology, combined with its exceptionally large, labiolingually broad conical teeth with ziphodont margins, indicates a capacity for generating strong bite forces capable of subduing sizable prey. Notably, the teeth of *Kostensuchus* gen. nov. are approximately twice the size of equivalent teeth in narrow-snouted peirosaurids, such as *Montealtosuchus* or *Hamadasuchus*, in both apicobasal height and mesiodistal length [see 58]. This anatomical evidence supports the interpretation that broad-snouted peirosaurids evolved into hypercarnivores by the end of the Cretaceous, increasing in body size and developing adaptations for hunting larger prey, likely medium-sized tetrapods such as ornithischian dinosaurs (based on the faunal assemblage of the Chorrillo Formation). *Kostensuchus* gen. nov. emerges as a large predator among its kin and within the Maastrichtian Chorrillo Formation, surpassed in size only by the megaraptorid theropod *Maip*. The discovery of *Kostensuchus* significantly enriches our understanding of the terrestrial ecosystems that developed along the floodplains of a freshwater setting under a temperate to warm, seasonally humid climate at high latitudes in Patagonia [12,14,15,59].

In the context of South America, broad-snouted peirosaurids are currently known from the latest Cretaceous at high-latitude regions, including the newly described *Kostensuchus* gen. nov. from the Chorrillo Formation and *Colhuehuapisuchus* from the Lago Colhue Huapi Formation [49]. These formations have similarities in their predator guild composition, with broad-snouted peirosaurids occupying a role alongside megaraptoran theropods as apex predators [18,60]. The growing understanding of vertebrate assemblages in these southern Patagonian ecosystems from the end of the Cretaceous highlights increasing differences compared to the better known faunas from other regions of South America, such as northern Patagonia (e.g., Allen Formation). In this region, predator guilds seem to have been dominated by abelisaurids, with no evidence of megaraptorans [5,18]. Additionally, large-bodied broad-snouted peirosaurids, such as

*Kostensuchus*, have not been identified in these areas despite decades of intensive sampling in northern Patagonia. The absence of megaraptorans and broad-snouted peirosaurids in northern Patagonia at the end of the Cretaceous might be indicative of distinct ecological settings compared to those in southern Patagonia. The causes behind these potential regional differences between northern and southern Patagonia remain poorly understood. Future research may shed light on whether these differences were driven by environmental factors, competitive interactions, or a combination of both.

## Broad-snouted peirosaurids and baurusuchids

The discovery of *Kostensuchus* and the completeness of its remains allow us to compare its anatomy with that of the other group of large predatory crocodyliforms from the Cretaceous of South America: Baurusuchidae [54,61,62]. These two clades of hypercarnivorous crocodyliforms share certain cranial and dental features related to their feeding habits, yet they also have structural differences in these anatomical regions. The dentition of *Kostensuchus* and baurusuchids share the presence of notably large, conical teeth that are almost as broad buccolingually as long mesiodistally, with small serrations along their entire mesial and distal margins. Both groups have a similar pattern of size variation in their dentition, including an enlarged premaxillary tooth, a hypertrophied maxillary tooth, and a similarly large fourth dentary tooth. While this pattern is not unique to these groups, the relative size of their large teeth (both in mesiodistal length and apicobasal height compared to skull size) is greater than in other notosuchians, with the exception of *Bretesuchus* and *Kaprosuchus* [63].

*Kostensuchus* gen. nov. and baurusuchids differ in several cranial and postcranial features. In the dentition, two notable differences are the presence of procumbent anterior dentary teeth in *Kostensuchus* gen. nov. (a trait not observed in baurusuchids or other peirosaurids) and a higher number of maxillary teeth, with *Kostensuchus* gen. nov. having over ten compared to the reduced number of five maxillary teeth in baurusuchids. In terms of skull proportions, baurusuchids (as well as many narrow-snouted peirosaurids) have a rostrum that is longer than the rest of the skull whereas *Kostensuchus* gen. nov. has a broad rostrum that is proportionately shorter. Additional differences in rostral morphology include the deeper and narrower proportions of the baurusuchid snout. This is especially noticeable in dorsal view, where the profile of the skull gradually tapers anteriorly in peirosaurids (including *Kostensuchus* gen. nov.), whereas in baurusuchids the snout lateral margins are parallel to each other and the skull broadens abruptly near the orbits. The anterior (symphyseal) region of the lower jaws in *Kostensuchus* gen. nov. is lateromedially broader and dorsoventrally lower compared to baurusuchids. Differences are also present in the posterior region of the mandibular ramus, where the adductor muscle attachments are located. In *Kostensuchus* gen. nov. and other peirosaurids, the external mandibular fenestra is small and low, and the angular has a low crest along its ventral margin. In baurusuchids, the external mandibular fenestra is larger, and the insertion of the *M. pterygoideus anterior* extends onto the lateral surface of the angular.

The well-preserved humerus of *Kostensuchus* gen. nov. allows for a detailed comparison with baurusuchids, revealing both similarities and differences in their forelimb anatomy. A notable similarity is the presence of a distomedially extended deltopectoral crest on the humerus of both *Kostensuchus* gen. nov. and baurusuchids. This shared feature has been interpreted as likely related to a parasagittal posture [40], but another possibility is that it reflects the presence of powerful forelimbs associated to prey capture or dismemberment during feeding. However, key differences are evident on the anterior surface of the distal humerus: *Kostensuchus* gen. nov. has a deep rounded fossa between the supracondylar crests and a well-developed latero-distal protuberance, features absent in baurusuchids. Furthermore, distal to this fossa, *Kostensuchus* gen. nov. bears a distinct step and a proximally facing shelf that extends lateromedially on the anterior surface of the humerus, above its distal condyles, a character shared with *Sebecus* [40] but absent in baurusuchids. The unique morphology of the humerus in *Kostensuchus* gen. nov. suggests potentially greater flexibility and a wider range of movement compared humerus of baurusuchids.

The ilium of *Kostensuchus* gen. nov. differs in many aspects from that of baurusuchids, in which the acetabulum is very deep and covered by a dorsoventrally thick, horizontally oriented acetabular roof (also present in *Notosuchus* and

*Sebecus* [40]). In contrast, the ilium of *Kostensuchus* gen. nov. has a shallower acetabulum, and the ventral surface of its roof faces ventrolaterally rather than ventrally, resembling the condition of uruguaysuchids [39]. Another difference in the ilium lies in the location of the insertion scars for the *M. iliotibialis*, responsible for hindlimb extension during the step cycle: in baurusuchids, these insertion areas are located at the external end of the laterally projected acetabular roof, while in *Kostensuchus* gen. nov. the insertion areas extend more posteriorly and dorsally and less laterally. The marked differences in the iliac morphology suggest locomotory and/or postural differences in the hip joint between these groups. In *Kostensuchus* gen. nov. the femur may have been positioned in a less erect or parasagittal orientation, with a more oblique line of action for the extensor muscles, compared to a more upright posture in baurusuchids. More remains of the hindlimb, especially of the femur, however would be needed to test this hypothesis.

These anatomical differences, particularly in the forelimb and pelvis, suggest that while both *Kostensuchus* gen. nov. and baurusuchids were large-bodied predators, their locomotor functions and ecological adaptations might have differed. This aligns with the contrasting lifestyles inferred for baurusuchids (terrestrial) and *Peirosaurus* [semiaquatic; 58,64]. Additional differences between *Kostensuchus* gen. nov. and baurusuchids, are also found in various anatomical regions, including the skull, dentition, axial skeleton, and dermal armor. These further emphasize the morphological and functional disparity of South American notosuchians, possibly due to their distinct phylogenetic histories, varied lifestyles, and feeding strategies, ultimately revealing the ecological diversity of large predatory crocodyliforms from the latest Cretaceous of South America.

Previous research has established that the plesiomorphic limb posture for crocodyliforms was parasagittal, a condition usually associated with terrestriality [65–69]. Within this framework, aquatic and semiaquatic habits are considered derived features that evolved along the lineage leading to extant crocodiles [65,68,70,71]. Notosuchians have generally been regarded as parasagittal and terrestrial forms [63,68,72–77], retaining this limb posture and associated habits from their basal crocodyliform ancestors [68]. The discovery of the notosuchian *Kostensuchus* gen. nov., with postcranial features indicative of a more sprawling limb posture, together with aquatic adaptations previously described for other peirosaurian lineages (i.e., mahajangasuchids, itasuchids [44,45,78–80], suggests that adaptations to a more aquatic lifestyle may have evolved more than once within peirosaurians and convergently with eusuchians.

## Conclusions

The discovery of *Kostensuchus atrox* gen. et sp. nov. considerably expands the knowledge about the anatomy of broad-snouted peirosaurids, previously known from extremely fragmentary remains from South America and Madagascar. *Kostensuchus* gen. nov. is retrieved as part of a clade of robust, broad-snouted peirosaurids that existed at the end of the Cretaceous across various regions of Gondwana. The new anatomical information provided by *Kostensuchus* gen. nov. sheds light on both, the similarities and differences between broad-snouted peirosaurids and baurusuchids, the other crocodyliform clade that independently evolved into apex predators during the Cretaceous of Gondwana.

*Kostensuchus* gen. nov. formed part of the latest Cretaceous ecosystem of southern Patagonia, in a freshwater ecosystem under a temperate to warm climate with seasonal humidity, alongside a diverse fauna of dinosaurs, mammals, and other vertebrates. The broad and high snout of *Kostensuchus* gen. nov., with notably large and robust ziphodont teeth, along with a broad adductor chamber in the skull and deep mandibular ramus, and robust forelimb anatomy suggests that the new species was capable of subduing large prey. These features imply that *Kostensuchus* gen. nov. played the role of a top predator within this end-Cretaceous ecosystem.

## Supporting information

**S1 File. Supplemental data.** Text with additional anatomical and phylogenetic datails and figures. (PDF)

**S2 File. Phylogenetic matrix.** Data matrix used in the phylogenetic study in TNT format.
(TNT)

## Acknowledgments

Authors would like to thank other members of the crew, including C. Sakata, C. Miyamae, H. Kamei, F. Brissón-Egli, A. Moreno, S. Miner, G. Muñoz, J. De Pasqua, C. Thompson, D. Piazza, G. Lo Coco, A. Misantone, and G. Stoll. A special thanks to Dr. Y. Harashi, General Director of the National Museum of Nature and Science, Japan. Special thanks to Santiago Miner for assistance in scanning and three-dimensional modeling of the fossils reported here. Thanks also to Federico Braun for allowing access to his property and Facundo Echeverría and his wife Daphne Fraser (Estancia La Anita) for their valuable help. We thank the Secretaría de Cultura for providing the permits to conduct our projects and explorations in Santa Cruz Province.

## Author contributions

**Conceptualization:** Fernando E. Novas, Diego Pol, Ismar de Souza Carvalho, Makoto Manabe, Takanobu Tsuihiji.

**Data curation:** Fernando E. Novas, Federico L. Agnolín, Makoto Manabe, Takanobu Tsuihiji, Sebastián Rozadilla, Gabriel L. Lio, Marcelo P. Isasi.

**Formal analysis:** Diego Pol.

**Methodology:** Diego Pol.

**Visualization:** Gabriel L. Lio.

**Writing – original draft:** Fernando E. Novas, Diego Pol.

**Writing – review & editing:** Fernando E. Novas, Diego Pol, Federico L. Agnolín, Ismar de Souza Carvalho, Makoto Manabe, Takanobu Tsuihiji, Sebastián Rozadilla, Gabriel L. Lio, Marcelo P. Isasi.

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
