## [Decision Letter · Decision Letter 0]

15 Apr 2025

PONE-D-25-04716A new large hypercarnivorous crocodyliform from the Maastrichtian of Southern Patagonia, ArgentinaPLOS ONE

Dear Dr. Pol,

Thank you for submitting your manuscript to PLOS ONE. After careful consideration, we feel that it has merit but does not fully meet PLOS ONE’s publication criteria as it currently stands. Therefore, we invite you to submit a revised version of the manuscript that addresses the points raised during the review process.

Please follow referee's suggestions mainly those about anatomical description and postcranial elementsOne referee asks, in partuclar, more adherence between descritptions and provided illustrations. A hypothetical life restoration could be of great benefit for the final publication.

We look forward to receiving your revised manuscript.

Kind regards,

Paolo Piras

Academic Editor

PLOS ONE

[Authors would like to thank other members of the crew, including C. Sakata, C. Miyamae, H. Kamei, F. Brissón-Egli, A. Moreno, S. Miner, G. Muñoz, J. De Pasqua, C. Thompson, D. Piazza, G. Lo Coco, A. Misantone, and G. Stoll. A special thanks to Dr. Y. Harashi, General Director of the National Museum of Nature and Science, Japan. Special thanks to Santiago Miner for assistance in scanning and threedimensional modeling of the fossils reported here. Thanks also to Federico Braun for allowing access to his property and Facundo Echeverría and his wife Daphne Fraser (Estancia La Anita) for their valuable help. We thank the Secretaría de Cultura for providing the permits to conduct our projects and explorations in Santa Cruz Province. This research is part of the Argentine-Japanese collaboration, and we thank funding provided by National Geographic Grant NGS 9282-R-22: The End of the Dinosaur Era in Patagonia, Fundação Carlos Chagas Filho de Amparo à Pesquisa do Estado do Rio de Janeiro (Faperj E-26/200.998/2024), Conselho Nacional de Desenvolvimento Científico e Tecnológico (CNPq 303596/2016-3).

[DP

9282-R-22

National Geographic Society

https://www.nationalgeographic.org/society/

ISC

Faperj E-26/200.998/2024

Fundação Carlos Chagas Filho de Amparo à Pesquisa do Estado do Rio de Janeiro

https://www.faperj.br/

ISC

CNPq 303596/2016-3

Conselho Nacional de Desenvolvimento Científico e Tecnológico

https://www.gov.br/cnpq/pt-br

The funders had no role in study design, data collection and analysis, decision to publish, or preparation of the manuscript.]

3. Please take this opportunity to be sure you have met all of our guidelines for new species. For proper registration of a new zoological taxon, we require two specific statements to be included in your manuscript.

1.        In the Results section, the globally unique identifier (GUID), currently in the form of a Life Science Identifier (LSID), should be listed under the new species name, for example:

Anochetus boltoni Fisher sp. nov. urn:lsid:zoobank.org:act:B6C072CF-1CA6-40C7-8396-534E91EF7FBB

Another LSID for the manuscript itself should also appear within the Nomenclature statement. You will need to contact Zoobank (zoobank.org/About) to obtain a GUID (LSID). You should receive one LSID for your manuscript and a separate, unique LSID for the new species.

2.        Please also insert the following text into the Methods section, in a sub-section to be called "Nomenclatural Acts":

The electronic edition of this article conforms to the requirements of the amended International Code of Zoological Nomenclature, and hence the new names contained herein are available under that Code from the electronic edition of this article. This published work and the nomenclatural acts it contains have been registered in ZooBank, the online registration system for the ICZN. The ZooBank LSIDs (Life Science Identifiers) can be resolved and the associated information viewed through any standard web browser by appending the LSID to the prefix "http://zoobank.org/". The LSID for this publication is: urn:lsid:zoobank.org:pub: XXXXXXX. The electronic edition of this work was published in a journal with an ISSN, and has been archived and is available from the following digital repositories: PubMed Central, LOCKSS [author to insert any additional repositories].

All PLOS ONE articles are deposited in PubMed Central and LOCKSS. If your institute, or those of your co-authors, has its own repository, we recommend that you also deposit the published online article there and include the name in your article.

Following a recent ruling by the International Commission on Zoological Nomenclature, electronic journals are now a valid format for publication of new zoological taxa. In order to ensure the valid publication of your new species, please be sure to include the updated version of Nomenclatural Acts (above). A complete explanation of our guidelines for publishing new species can be found on our website: http://www.plosone.org/static/guidelines#zoological .

4. We are unable to open your Figure file [Fig 11-textoutline.eps]. Please kindly revise as necessary and re-upload.

Reviewers' comments:

Reviewer's Responses to Questions

**Comments to the Author**

1. Is the manuscript technically sound, and do the data support the conclusions?

Reviewer #1: Yes

Reviewer #2: Yes

2. Has the statistical analysis been performed appropriately and rigorously?

Reviewer #1: Yes

Reviewer #2: N/A

3. Have the authors made all data underlying the findings in their manuscript fully available?

Reviewer #1: Yes

Reviewer #2: Yes

4. Is the manuscript presented in an intelligible fashion and written in standard English?

Reviewer #1: Yes

Reviewer #2: Yes

5. Review Comments to the Author

Reviewer #1: The work is very well written and developed, and the specimen itself is extraordinary for the understanding of Notosuchia, without a doubt.

However, some corrections and modifications are necessary to improve the work.

I believe that future work focused on the taphonomic aspects of the material will be carried out. However, given the interesting fact that it was found in a concretion, it would be appropriate to report part of the process of discovery, collection and preparation of the specimen, even if briefly.

In terms of the morphological description, I felt that more attention should have been paid to the palatal region of the skull, and its elements such as palatines, pterygoid, and ectopterygoids.

As for the postcranium, the osteoderms (dorsal dermal shield) have practically not been described.

In the discussions, when commenting on the differences in the paleofaunas of the southern and northern Patagonia and Brazil, it should be noted that Barreirosuchus franciscoi is also a broad-snouted taxon of Peirosauria with large body mass. There is also evidence of megaraptors in the southeastern region of Brazil.

My comments and suggested changes are highlighted and are included in the attached files.

Reviewer #2: I was requested to review the manuscript titled “A new large hypercarnivorous crocodyliform from the Maastrichtian of Southern Patagonia, Argentina” by Novas et al. The manuscript deals with a beautifully preserved new taxon (Kostensuchus) from the latest Cretaceous (Maastrichtian) of Southern Argentina. Besides the good preservation, this new taxon is very large and helps in the knowledge to a clade only known by very partial remains, “broad-snouted” peirosaurids. Furthermore, illustrations of the fossil materials are adequate and contribute to illustrate the specimen.

The manuscript is clear and well-written. I made some comments (typos and some incomplete sentences), but those should be considered as suggestions by the authors as I am not a native English speaker. I think after some minor changes the manuscript could be accepted for publication. I mention, briefly some of these (which are minor):

1) Figures should be cited more often in the description. I added some cites to this, but only in the begging.

2) Some cites are missing when comparing the anatomy of Kostensuchus. E.g., “…unlike the condition of most notosuchians”. That should be followed by a cite or at least the mention of some taxa. See the comment (cites) or (e.g., MENTION SOME TAXA) throughout the text.

3) Some particular, even unique features, of Kostensuchus can not be seen in the figures (like the grooves on the premaxilla, mandible, or the lack/change of ornamentation near the antorbital fenestra). I don’t know if that is caused by the poor quality of images, if so label them on the figures. Regarding the unique condition of the lacrimal of Kostensuchus, that could be illustrated in a very simple one-column figure (especially considering it represent an autapomorphy of the new taxon).

4) Add more labels to the figures, in particular to the ones of the posterior view of the skull (some features are described but not marked like the siphonium) and the cervicodorsal block of vertebrae. In the later, if possible add the number of vertebrae (either D1, D2, D3) as it is useful to evaluate some transitions.

5) Finally, a small comment on the humeral anatomy of Kostensuchus. The authors infer a more sprawiling limb orientation during the locomotion of Kostensuchus, mostly based on the anatomy of the ilium. However, some of these features were used as criteria to infer a more parasagittal orientation of the limbs in sebecids and baurusuchians (Riff, 2007; Pol et al., 2012). Contrasting with those taxa, Kostensuchus has a much more robust humerus, which is not common un cursorial parasagittal reptiles. Thus, these features could have other putative functions here (i.e., supporting the weight if the anterior part of the body and head?)?

6. PLOS authors have the option to publish the peer review history of their article (what does this mean? ). If published, this will include your full peer review and any attached files.

**Do you want your identity to be public for this peer review?**  For information about this choice, including consent withdrawal, please see our Privacy Policy .

Reviewer #1: No

Reviewer #2: **Yes: ** Juan Martín Leardi

---

## [Author Response · Author response to Decision Letter 1]

25 Jun 2025

Dear Editor,

We thank the reviewers and the editor for their thorough evaluation of our manuscript. We have followed all suggestions except those for which we provide detailed response below in bold font:

Reviewer #1:

The work is very well written and developed, and the specimen itself is extraordinary for the understanding of Notosuchia, without a doubt.

However, some corrections and modifications are necessary to improve the work.

I believe that future work focused on the taphonomic aspects of the material will be carried out. However, given the interesting fact that it was found in a concretion, it would be appropriate to report part of the process of discovery, collection and preparation of the specimen, even if briefly.

Whereas conducting a taphonomic research could be interesting, it lies outside the scope of this work and especially outside our expertise, as we do not have authors capable of conducting a proper taphonomic analysis. However, we follow the advice of the reviewer and we have now included a report on the process of discovery, collection and preparation of the specimen. Given the structure of the manuscript we have decided to include these details in the supplementary information.

In terms of the morphological description, I felt that more attention should have been paid to the palatal region of the skull, and its elements such as palatines, pterygoid, and ectopterygoids.

We have expanded the description of the palate, especially of the palatines and pterygoids that are better exposed than the ectopterygoids.

As for the postcranium, the osteoderms (dorsal dermal shield) have practically not been described.

We have included a new paragraph description of the osteoderms

In the discussions, when commenting on the differences in the paleofaunas of the southern and northern Patagonia and Brazil, it should be noted that Barreirosuchus franciscoi is also a broad-snouted taxon of Peirosauria with large body mass.

There is also evidence of megaraptors in the southeastern region of Brazil.

Yes, good points, we have modified the text to emphasize the differences with northern Patagonia rather than with Brazil. Restricting the comparison to northern Patagonia makes also more sense because there is chronostratigraphic control that correlate the Chorrillo and the Allen formations, wheras the correlation with the Baurú Group of Brazil is less precise.

My comments and suggested changes are highlighted and are included in the attached files.

All changes in the marked-up file have been followed.

Reviewer #2:

I was requested to review the manuscript titled “A new large hypercarnivorous crocodyliform from the Maastrichtian of Southern Patagonia, Argentina” by Novas et al. The manuscript deals with a beautifully preserved new taxon (Kostensuchus) from the latest Cretaceous (Maastrichtian) of Southern Argentina. Besides the good preservation, this new taxon is very large and helps in the knowledge to a clade only known by very partial remains, “broad-snouted” peirosaurids. Furthermore, illustrations of the fossil materials are adequate and contribute to illustrate the specimen.

The manuscript is clear and well-written. I made some comments (typos and some incomplete sentences), but those should be considered as suggestions by the authors as I am not a native English speaker.

All changes in the marked up file have been followed.

I think after some minor changes the manuscript could be accepted for publication. I mention, briefly some of these (which are minor):

1) Figures should be cited more often in the description. I added some cites to this, but only in the begging.

Corrected, we have added more references to figures in the description.

2) Some cites are missing when comparing the anatomy of Kostensuchus. E.g., “…unlike the condition of most notosuchians”. That should be followed by a cite or at least the mention of some taxa. See the comment (cites) or (e.g., MENTION SOME TAXA) throughout the text.

Corrected.

3) Some particular, even unique features, of Kostensuchus can not be seen in the figures (like the grooves on the premaxilla, mandible, or the lack/change of ornamentation near the antorbital fenestra). I don’t know if that is caused by the poor quality of images, if so label them on the figures.

We have added a new figure (Figure 4) with anatomical details, see below.

Regarding the unique condition of the lacrimal of Kostensuchus, that could be illustrated in a very simple one-column figure (especially considering it represent an autapomorphy of the new taxon).

We have added a new figure (Figure 4) with anatomical details that were not well illustrated in the original figures. These include the maxillary groove, the lacrimal bulge, the autapomorphic depressed lateral surface of lacrimal, the lack of lacrimal antorbital fossa, and the extension of the jugal towards the antorbital fenestra.

4) Add more labels to the figures, in particular to the ones of the posterior view of the skull (some features are described but not marked like the siphonium) and the cervicodorsal block of vertebrae. In the later, if possible add the number of vertebrae (either D1, D2, D3) as it is useful to evaluate some transitions.

Corrected.

5) Finally, a small comment on the humeral anatomy of Kostensuchus. The authors infer a more sprawiling limb orientation during the locomotion of Kostensuchus, mostly based on the anatomy of the ilium. However, some of these features were used as criteria to infer a more parasagittal orientation of the limbs in sebecids and baurusuchians (Riff, 2007; Pol et al., 2012). Contrasting with those taxa, Kostensuchus has a much more robust humerus, which is not common un cursorial parasagittal reptiles. Thus, these features could have other putative functions here (i.e., supporting the weight if the anterior part of the body and head?)?

We have revised the relevant sentence to reflect a more nuanced interpretation. It now reads:

“This shared feature has been interpreted as likely related to a parasagittal posture [40], but another possibility is that it reflects the presence of powerful forelimbs associated with prey capture or dismemberment during feeding.”

We also added a sentence in the ilium discussion, where we discuss the possible orientation of the hindlimb to acknowledge the limitations of the current data and the need for further evidence:

“More remains of the hindlimb, especially of the femur, however, would be needed to test this hypothesis.”

Figure 9:You mention this distolateral feature. Label it

We have labeled the anterior distal fossa already.

Thank you for considering our manuscript. We look forward to the opportunity to share our findings with the scientific community through PLOS ONE. Please do not hesitate to contact me if additional information is required.

Sincerely,

Diego Pol

Museo Argentino de Ciencias Naturales “Bernardino Rivadavia” (MACN)

Buenos Aires, Argentina

---

## [Editor Report · Decision Letter 1]

3 Jul 2025

A new large hypercarnivorous crocodyliform from the Maastrichtian of Southern Patagonia, Argentina

PONE-D-25-04716R1

Dear Dr. %Pol%,

We’re pleased to inform you that your manuscript has been judged scientifically suitable for publication and will be formally accepted for publication once it meets all outstanding technical requirements.

Kind regards,

Paolo Piras

Academic Editor

PLOS ONE
---

## [Editor Report · Acceptance letter]

PONE-D-25-04716R1

PLOS ONE

Dear Dr. Pol,

I'm pleased to inform you that your manuscript has been deemed suitable for publication in PLOS ONE. Congratulations! Your manuscript is now being handed over to our production team.

Kind regards,

on behalf of

Dr. Paolo Piras

Academic Editor

PLOS ONE